# ViDA: Homeostatic Visual Domain Adapter for Continual Test Time Adaptation

**Jiaming Liu**[1], **Senqiao Yang**[1*], **Peidong Jia** [1†], **Renrui Zhang** [3],

Ming Lu[1], Yandong Guo[2], Wei Xue[4], Shanghang Zhang[1 ✉]

[1]National Key Laboratory for Multimedia Information Processing,
School of Computer Science, Peking University [2]AI[2]Robotics
[3] The Chinese University of Hong Kong [4] Hong Kong University of Science and Technology
jiamingliu@stu.pku.edu.cn, yangsenqiao.ai@gmail.com, shanghang@pku.edu.cn

## ABSTRACT

Since real-world machine systems are running in non-stationary environments, Continual Test-Time Adaptation (CTTA) task is proposed to adapt the pre-trained model to continually changing target domains. Recently, existing methods mainly focus on model-based adaptation, which aims to leverage a self-training manner to extract the target domain knowledge. However, pseudo labels can be noisy and the updated model parameters are unreliable under dynamic data distributions, leading to error accumulation and catastrophic forgetting in the continual adaptation process. To tackle these challenges and maintain the model plasticity, we design a Visual Domain Adapter (ViDA) for CTTA, explicitly handling both domain-specific and domain-shared knowledge. Specifically, we first comprehensively explore the different domain representations of the adapters with trainable high-rank or low-rank embedding spaces. Then we inject ViDAs into the pre-trained model, which leverages high-rank and low-rank features to adapt the current domain distribution and maintain the continual domain-shared knowledge, respectively. To exploit the low-rank and high-rank ViDAs more effectively, we further propose a Homeostatic Knowledge Allotment (HKA) strategy, which adaptively combines different knowledge from each ViDA. Extensive experiments conducted on four widely used benchmarks demonstrate that our proposed method achieves state-of-the-art performance in both classification and segmentation CTTA tasks. Note that, our method can be regarded as a novel transfer paradigm for large-scale models, delivering promising results in adaptation to continually changing distributions.

## 1 INTRODUCTION

Deep Neural Networks (DNN) have achieved remarkable performance in various computer vision tasks, such as classification (He et al., 2016), object detection (Zhu et al., 2020), and segmentation (Xie et al., 2021) when the test data distribution is similar to the training data. However, real-world machine perception systems (Yang et al., 2023a; Li et al., 2023; Arnold et al., 2019) operate in non-stationary and constantly changing environments, which contain heterogeneous and dynamic domain distribution shifts. Applying a pre-trained model in these real-world tasks (Sakaridis et al., 2021) can lead to significant degradation in perception ability on target domains, especially when the target distribution changes unexpectedly over time. Therefore, the development of continual domain adaptation (DA) methods is essential for enhancing the generalization capability of DNNs and improving the reliability of machine perception systems in dynamic environments.

A classical source-free DA task, Test-Time Adaptation (Liang et al., 2023) (TTA), eases the distribution shift between a source domain and a fixed target domain. This is typically achieved through the utilization of self-training mechanisms (Mummadi et al., 2021; Wang et al., 2020). However, when adapting to continually changing target domains, pseudo labels are noisy and the updated

---

*Equal contribution, † Equal technical contribution,
✉ Corresponding author. Project page: https://sites.google.com/view/iclr2024-vida/home

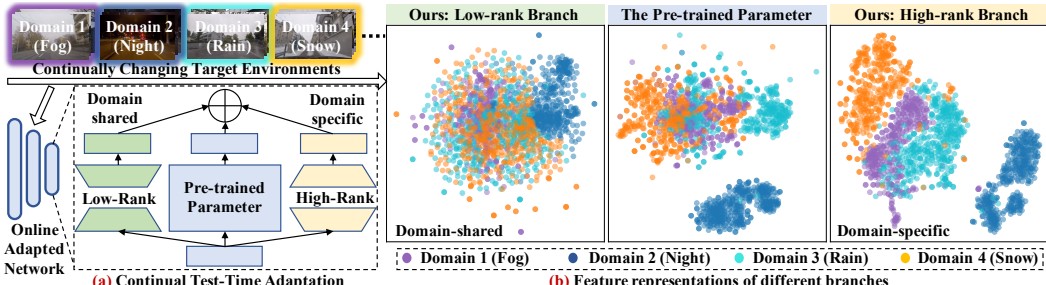

Figure 1: **The problem and motivation.** (a) Our goal is to effectively adapt the source pre-trained model to continually changing target domains. We propose Visual Domain Adapters with high-rank and low-rank embedding spaces to tackle the error accumulation and catastrophic forgetting challenges during the continual adaptation process. (b) we conduct a t-SNE (Van der Maaten & Hinton, 2008) distribution analysis for the different adapter representations across four target domains (ACDC). The low-rank branch exhibits a consistent distribution across the target domains, suggesting that it can effectively disregard the impact of dynamic distribution shifts. The high-rank branch demonstrates noticeable distribution discrepancies between the various target domains, suggesting that it primarily focuses on extracting domain-specific knowledge.

model parameters become uncertain, leading to error accumulation and catastrophic forgetting. To tackle this problem, Continual Test-Time Adaptation (CTTA) has been proposed (Wang et al., 2022), which addresses a sequence of different distribution shifts over time rather than a single shift as in TTA. Furthermore, CTTA also encompasses the efficient continual adaptation of foundation models (Kirillov et al., 2023) to continual downstream tasks or distributions (Bahng et al., 2022).

Existing CTTA works have primarily employed model-based or prompt-based approaches to extract target domain-specific and domain-shared knowledge simultaneously. However, for model-based methods (Wang et al., 2022; Chakrabarty et al., 2023), the noisy pseudo labels are still unreliable and play a limited role in avoiding error accumulation, particularly in scenarios with significant distribution gaps. Meanwhile, prompt-based methods (Gan et al., 2023; Yang et al., 2023b) face difficulties in leveraging soft prompts with limited trainable parameters to learn long-term domain-shared knowledge and prevent catastrophic forgetting.

To tackle these limitations and maintain the model plasticity, we design a homeostatic Visual Domain Adapter (ViDA), shown in Fig .1 (a), which explicitly manages domain-specific and domain-shared knowledge in the continual adaptation process. Specifically, we first carefully explore the different domain representations of ViDAs with trainable high or low-dimension embedding space in the middle layer. As shown in Fig. 1 (b), our observations reveal that ViDA with a low-rank embedding space focuses on task-relevant feature representation, showing trivial distribution distance in different domains and neglecting the influence of dynamic distribution shifts. Conversely, ViDA with a high-rank feature concentrates more on extracting domain-specific knowledge, as evidenced by the feature distribution in different target domains showing an obvious discrepancy. We provide a detailed explanation of the motivations in Section 3.1 and Appendix B.

This observation motivates us to inject ViDAs into the pre-trained model, which leverages different domain representations of high and low-dimension features to avoid error accumulation and catastrophic forgetting simultaneously. To better extract different domain knowledge, we further propose a Homeostatic Knowledge Allotment (HKA) strategy to dynamically fuse the knowledge from low-rank and high-rank ViDA. Based on the data distribution, HKA adaptively regularizes the balance of different feature representations, including original model, domain-specific, and task-relevant features. During inference, the low-rank and high-rank ViDAs can be projected into the pre-trained model by re-parameterization (Ding et al., 2021), which ensures no extra parameter increase and maintains the model plasticity. In summary, our contributions are as follows:

**1)** We study the different domain representations of the adapters with high-rank and low-rank features. Then we design a Visual Domain Adapter (ViDA), explicitly managing domain-specific and task-relevant knowledge to tackle the error accumulation and catastrophic forgetting problem, respectively.

**2)** Considering the various distribution shifts for each target sample, we further propose a Homeostatic Knowledge Allotment (HKA) strategy to dynamically merge knowledge from low-rank and high-rank ViDAs, thus enhancing ViDAs' distinct domain representations.

**3)** Our CTTA method provides a novel transfer paradigm for large-scale models, delivering promising results in adaptation to continually changing distributions. Meanwhile, we empower the source model with domain generalization ability through the proposed homeostatic ViDAs, achieving a significant improvement on the unseen target domains.

## 2 RELATED WORK

**Test-time adaptation (TTA)**, also referred to as source-free domain adaptation (Boudiaf et al., 2023; Kundu et al., 2020; Yang et al., 2021), aims to adapt a source model to an unknown target domain distribution without relying on any source domain data. Recent research has explored self-training and entropy regularization techniques to fine-tune the source model (Liang et al., 2020; Chen et al., 2022). Tent (Wang et al., 2021) updates the training parameters in batch normalization layers by minimizing entropy. This approach has prompted subsequent exploration in recent works (Niu et al., 2023; Yuan et al., 2023), which continue to investigate the robustness of normalization layers.

**Continual Test-Time Adaptation (CTTA)** refers to a scenario where the target domain is not static, presenting additional challenges for traditional TTA methods. The first approach to address this challenging task is introduced in (Wang et al., 2022), which combines bi-average pseudo labels and stochastic weight reset. For addressing error accumulation, Ecotta (Song et al., 2023) introduces a meta-network to regularize the outputs from both the meta-network and the frozen network. And RMT (Döbler et al., 2023) introduces a symmetric cross-entropy loss. While these works tackle the CTTA problem at the model level, (Gan et al., 2023; Yang et al., 2023b; Ni et al., 2023) utilize visual domain prompts or a small fraction of parameters to extract continual target domain knowledge.

**Parameter-Efficient Fine-Tuning (PEFT)** has gained significant traction within the field of natural language processing (NLP) (Hu et al., 2021; Houlsby et al., 2019; Zaken et al., 2021; Hu et al., 2022; Gao et al., 2021; He et al., 2021; Vu et al., 2022; Qin et al., 2021). Adapter-based models, a form of PEFT, have gained popularity in NLP. They employ bottleneck architecture adapter modules inserted between layers in pre-trained models. Inspired by NLP, adapters in visual tasks have also received widespread attention. In the initial phases of adapter development, residual adapter modules (Rebuffi et al., 2017; 2018) are proposed to aid in the effective adaptation of convolutional neural networks across multiple downstream tasks. AdaptFormer (Chen et al.) enhances the ViT (Dosovitskiy et al., 2020) model by replacing the original multi-layer perceptron (MLP) block with a down-to-up bottleneck module in a parallel manner. VL-Adapter (Sung et al., 2022) improves the efficiency and performance of adapters by sharing low-dimensional layer weights to attain knowledge across tasks.

## 3 METHOD

**Preliminary.** In Continual Test-Time Adaptation (CTTA), we pre-train the model $q_\theta(y|x)$ on the source domain $D_S = (Y_S, X_S)$ and adapt it on multiple target domains $D_{T_i} = \{(X_{T_i})\}_{i=1}^n$, where $n$ represents the scale of the continual target datasets. The entire process can not access any source domain data and can only access target domain data once. The distributions of the target domains (i.e., $D_{T_1}, D_{T_2}, ..., D_{T_n}$) are constantly changing over time. Our goal is to adapt the pre-trained model to target domains and maintain the perception ability of the model on the seen domain distributions.

**Overall Framework.** Drawing from the insight that mean teacher predictions are often more robust than standard models (Tarvainen & Valpola, 2017b; Döbler et al., 2023), we utilize a teacher-student framework to ensure stability during continual domain adaptation, which also presents a fair comparison with previous CTTA works (Wang et al., 2022; Gan et al., 2022). The overall framework and the details of our method are shown in Fig .2.

### 3.1 MOTIVATION

The CTTA encounters significant challenges, primarily due to error accumulation and catastrophic forgetting (Wang et al., 2022). Meanwhile, adapters with different dimensional middle-layer features demonstrate effectiveness in addressing these challenges. This encourages us to take a step further and justify the principles underlying the use of low-rank adapter and high-rank adapter in the CTTA.

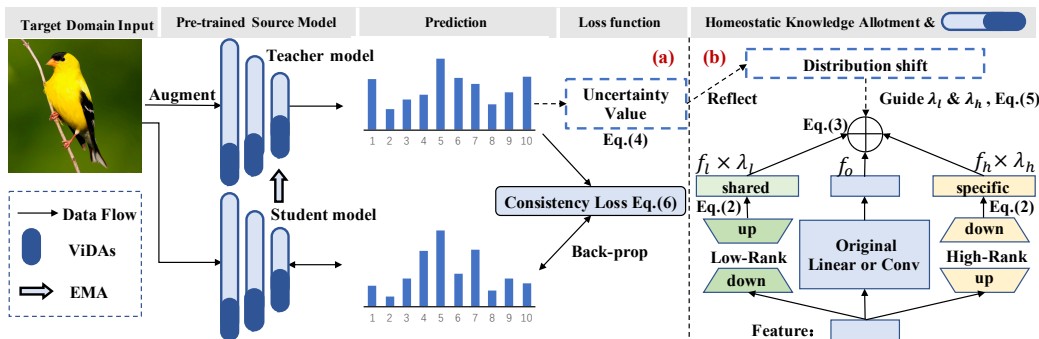

Figure 2: **The framework of Visual Domain Adapter (ViDA).** (a) We inject low-rank and high-rank ViDAs into either linear or Conv layers of the pre-trained source model. The student model processes the original image, while the teacher model processes an augmented version of the same image. To update the ViDAs, we construct a teacher-student framework and use a consistency loss (Eq. 6) as the optimization objective. In addition, the teacher model calculates an uncertainty value (Eq. 4), reflecting the distribution shift of each sample in target domains. (b) Based on the degree of distribution shift, we introduce the Homeostatic Knowledge Allotment (HKA) strategy, which aims to dynamically fuse the knowledge from each ViDA with different domain representation.

**Low-rank adapter.** Our hypothesis regarding the effectiveness of adapters in mitigating catastrophic forgetting is that their low-rank embedding space representation plays a crucial role. To explore this further, we conduct a t-SNE distribution study (Van der Maaten & Hinton, 2008) on the third transformer block to analyze the feature distributions across four target domains (ACDC dataset (Sakaridis et al., 2021)). The results are depicted in Fig. 1 (b). Our analysis reveals that the low-rank adapter exhibits a relatively consistent distribution across the different target domains, suggesting that its low-rank embedding space can effectively disregard the impact of dynamic distribution shifts and prioritize the extraction of domain-shared knowledge.

Furthermore, we adopt the domain distance definition proposed by Ben-David (Ben-David et al., 2006; 2010) and build upon previous domain transfer research (Ganin et al., 2016) by employing the $\mathcal{H}\text{-}divergence$ metric to evaluate the domain representations of adapters across different target domains. The discrepancy distance between two distributions $D_S$ and $D_{T_i}$ can be calculated as:

$$d_{\mathcal{H}}(D_S, D_{T_i}) = 2 \sup_{\mathcal{D} \sim \mathcal{H}} | \Pr_{x \sim D_S} [\mathcal{D}(x) = 1] - \Pr_{x \sim D_{T_i}} [\mathcal{D}(x) = 1]| \qquad (1)$$

, where $\mathcal{H}$ denotes hypothetical space and $\mathcal{D}$ denotes discriminator. Similar to (Ruder & Plank, 2017; Allaway et al., 2021), we adopt the $Jensen\text{-}Shannon\ (JS)\ divergence$ between two adjacent domains as an approximation of $\mathcal{H}$-divergence because it has been shown to successfully distinguish domains. If the inter-domain divergence is relatively small, it can be demonstrated that the feature representation is consistent and less influenced by cross-domain shifts (Ganin et al., 2016). We compare the $JS$ values obtained by using the source model alone, injecting a low-rank adapter, injecting a high-rank adapter, and combining low- and high-rank adapters, as illustrated in Fig. 3 (a). Our results indicate that the feature representation generated by the low-rank adapter exhibits lower divergence compared to both the original source model and the high-rank adapter, especially when dealing with later target domains or significant domain shifts between adjacent domains (i.e., target domains 9-13). This result simultaneously demonstrates the low-rank adapter's ability to learn long-term domain-shared knowledge in a continually changing environment.

To provide clearer evidence for the intuition, we extend our analysis by incorporating the qualitative analysis of Class Activation Mapping (CAM) on the ImageNet-to-ImageNet-C CTTA. As shown in Fig .4, we showcase the feature representations from different target domains, including the noise of Gaussian and Snow. We observe that the low-rank ViDA is inclined to put more weight on the foreground sample while tending to disregard background noise shifts. This indicates that the low-rank ViDA attends to locations with more general and task-relevant information.

**High-rank adapter.** Regarding the domain representation of the adapter with a high-rank feature, we propose that it is better suited to address error accumulation in the continual adaptation process. We verify this by t-SNE analyzing the feature distributions between different domains, as shown in Fig. 1 (b), and observe that there is a clear discrepancy between domains. The distribution achieves a better aggregation in a single domain. This suggests that high-ranking adapters have a

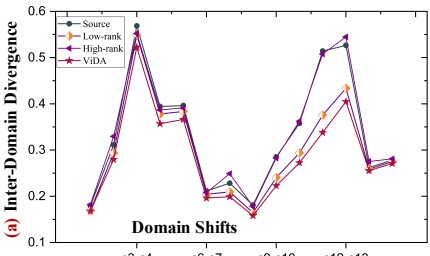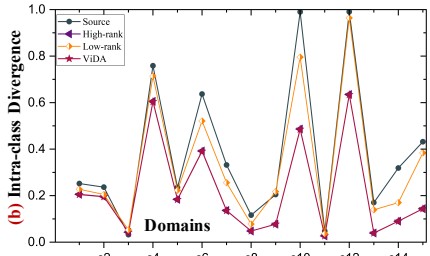

Figure 3: c1 to c15 represent the 15 corruption domains in CIFAR10C listed in sequential order. (a) Low-rank adapter based model effectively mitigates inter-domain divergence than the source model across all 14 domain shifts. (b) High-rank adapter based model significantly enhances the intra-class feature aggregation, yielding results that closely approximate those achieved by our ViDA method.

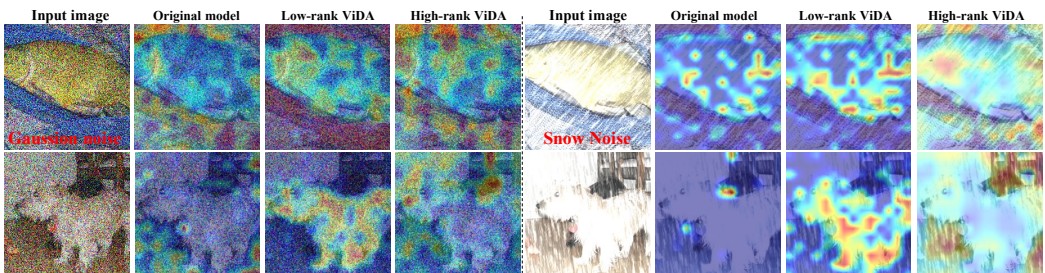

Figure 4: The qualitative analysis of the CAM. We adopt CAM to compare the attention of the low-rank branch, high-rank branch, and the original model during the continual adaptation process.

better grasp of target domain data distribution. Inspired by intra-class dissimilarity proposed by $k$-means (MacQueen, 1967), we use normalized intra-class divergence to further verify the domain representations of high-rank adapters in CIFAR10C. In a given domain, if the intra-class divergence for each category is smaller, it demonstrates that the model has a better understanding of the current distribution (Li et al., 2020). As illustrated in Fig. 3 (b), the high-rank adapter is found to drive down the intra-class divergence within almost all domains, indicating that it can better adapt to current domain distribution and extract domain-specific knowledge in continual target domains. For more straightforward verification, we conduct qualitative analysis by incorporating the visualization of CAM. Conversely, the high-rank ViDA exhibits an inverse pattern, as illustrated in Fig .4. It allocates more attention to locations characterized by substantial domain shift, encompassing the entirety of the input images. This behavior aligns with the high-rank branch's tendency to fit global information and predominantly extract domain-specific knowledge from the target domain data.

In conclusion, the structure of low-rank ViDA reduces feature redundancy, which leads to an underfit state during CTTA. Consequently, it tends to acquire general information across continuous target domains, extracting task-relevant knowledge to mitigate catastrophic forgetting. In contrast, high-rank ViDA employs a higher-dimensional feature representation that better aligns with the target data distribution, thereby focusing on learning domain-specific knowledge to prevent error accumulation. We offer additional justifications and specifically designed experiments in Appendix B.

## 3.2 VISUAL DOMAIN ADAPTER

The above observation motivates us to introduce high-rank and low-rank Visual Domain Adapters (ViDAs) into the source pre-trained model, aiming to simultaneously adapt current domain distribution and maintain the continual domain-shared knowledge in CTTA.

**The architecture.** The design principle of injecting ViDAs into the pre-trained model is simple yet effective, which is illustrated in Fig .2 (b). As we can see there are three sub-branches, the linear (or Conv) layer in the middle branch is originated from the original network, while the right branch and left branch are bottleneck structures and separately indicate the high-rank ViDA and low-rank ViDA. Specifically, the right branch (high-rank) contains an up-projection layer with parameters $W_{up}^h \in R^{d \times d_h}$, a down-projection layer with parameters $W_{down}^h \in R^{d_h \times d}$, where $d_h$ (e.g., $d_h = 128$) is the middle dimension of high-rank feature and satisfies $d_h \geq d$. There is not any non-linear layer

in the ViDA. And we utilize the linear layer as the projection layer when the original model is transformer architecture and adopt $1 \times 1$ Conv as the projection layer when the original model is a convolution network. In contrast, the left low-rank branch first injects a down-projection layer with parameters $W_{down}^l \in R^{d \times d^l}$, then place an up-projection layer with parameters $W_{up}^l \in R^{d_l \times d}$, where $d_l$ (e.g., $d_l = 1$) stand for the middle dimension of the low-rank feature ($d_l \ll d$). For a input feature $f$, the produced features of high-rank ($f_h$) and low-rank ViDA ($f_l$) are formulated as:

$$f_h = W_{down}^h \cdot (W_{up}^h \cdot f); \quad f_l = W_{up}^l \cdot (W_{down}^l \cdot f) \tag{2}$$

The two-branch bottleneck is connected to the output feature of the original network ($f_o$) through the residual connection via scale factors ($\lambda_h$ and $\lambda_l$). The fusion knowledge ($f_f$) can be described as:

$$f_f = f_o + \lambda_h \times f_h + \lambda_l \times f_l \tag{3}$$

The domain knowledge scale factors ($\lambda_h$ and $\lambda_l$) are adaptively obtained through the homeostatic knowledge allotment strategy, which is shown in Section 3.3. During inference, the different domain-represented ViDAs (linear relation) can be projected into the pre-trained model by re-parameterization (Ding et al., 2021), which ensures no extra model parameter increase of the original model.

### 3.3 HOMEOSTATIC KNOWLEDGE ALLOTMENT

**Method motivation.** In CTTA, target domain data can only be accessed once and exhibits different distribution shifts, which underscores the importance of efficient domain transfer. Moreover, to effectively address error accumulation and catastrophic forgetting, it becomes necessary to extract different knowledge and manage it separately. Although the specialized structures of low-rank and high-rank ViDAs contribute to distinct domain representation learning, the continual adaptation process also needs to regularize the knowledge fusion weight to ensure the efficient capture of relevant domain-specific knowledge without compromising the retention of long-term domain-shared knowledge. **HKA design.** As depicted in Fig .2 (b), we draw inspiration from (Ovadia et al., 2019; Roy et al., 2022) and introduce an uncertainty value to quantify the degree of distribution shift for each sample. While the confidence score is a common measure to assess prediction reliability, it tends to fluctuate irregularly and becomes unreliable in continual changing environment. To address this limitation, we employ the MC Dropout technique (Gal & Ghahramani, 2016) on linear layers, enabling multiple forward propagations to obtain $m$ sets of probabilities for each sample. Subsequently, we calculate the uncertainty value $\mathcal{U}(x)$ for a given input $x$, which are formulated as:

$$\mathcal{U}(x) = \left( \frac{1}{m} \sum_{i=1}^{m} \|p_i(y|x) - \mu\|^2 \right)^{\frac{1}{2}} \tag{4}$$

Where $p_i(y|x)$ is the predicted probability of the input $x$ in the $i^{th}$ forward propagation and $\mu$ is the average value of $m$ times prediction. To dynamically adjust the scale factors ($\lambda_h$ and $\lambda_l$) based on the uncertainty score, the formulation is as follows:

$$\begin{cases} \lambda_h = 1 + \mathcal{U}(x) & \lambda_l = 1 - \mathcal{U}(x), & \mathcal{U}(x) \geq \Theta \\ \lambda_h = 1 - \mathcal{U}(x) & \lambda_l = 1 + \mathcal{U}(x), & \mathcal{U}(x) < \Theta \end{cases} \tag{5}$$

The threshold value of uncertainty is denoted as $\Theta$, where $\Theta = 0.2$. To realize the homeostasis of different domain knowledge, when facing the sample with a large uncertainty value, we adaptively increase the fusion weight of domain-specific knowledge ($\lambda_h$). Conversely, if the input has a low uncertainty value, the fusion weight of domain-shared knowledge ($\lambda_l$) will be increased.

### 3.4 OPTIMIZATION OBJECTIVE

Following previous CTTA work (Wang et al., 2022), we leverage the teacher model $\mathcal{T}$ to generate the pseudo labels $\widetilde{y}$ for updating ViDAs. And the consistency loss $L_{ce}$ is the optimization objective.

$$\mathcal{L}_{ce}(x) = -\frac{1}{C} \sum_{c}^{C} \widetilde{y}(c) \log \hat{y}(c) \tag{6}$$

Where $\hat{y}$ is the output of our student model $\mathcal{S}$, $C$ means the number of categories. Same as previous works(Gan et al., 2023), we load the source pre-trained parameters to initialize the weight of both models and adopt the exponential moving average (EMA) to update the teacher model with ViDAs.

$$\mathcal{T}^t = \alpha \mathcal{T}^{t-1} + (1 - \alpha)\mathcal{S}^t \tag{7}$$

Where $t$ is the time step. And we set the updating weight $\alpha = 0.999$ (Tarvainen & Valpola, 2017a).

# 4 EXPERIMENT

In Section 4.2 and 4.3, we compare our method with other SOTA methods on classification and semantic segmentation CTTA. In Section 4.4, we employ the foundation models (DINOv2 (Kirillov et al., 2023) and SAM (Oquab et al., 2023)) as the backbone and evaluate the efficacy of our method. In Section 4.5, we further evaluate the domain generalization ability of the proposed method on unseen target domains. Comprehensive ablation studies are conducted in Section 4.6. More quantitative comparisons and qualitative analyses are shown in the Appendix C and D, respectively.

## 4.1 TASK SETTINGS AND DATASETS

**Dataset.** We evaluate our method on three classification CTTA benchmarks, including CIFAR10-to-CIFAR10C, CIFAR100-to-CIFAR100C (Krizhevsky et al., 2009) and ImageNet-to-ImageNet-C (Hendrycks & Dietterich, 2019). For segmentation CTTA (Yang et al., 2023b), we evaluate our method on Cityscapes-to-ACDC, where the Cityscapes dataset (Cordts et al., 2016) serves as the source domain, and the ACDC dataset (Sakaridis et al., 2021) represents the target domains.

**CTTA Task setting.** Following (Wang et al., 2022), in classification CTTA tasks, we sequentially adapt the pre-trained source model to the fifteen target domains with the largest corruption severity (level 5). The online prediction results were evaluated immediately after encountering the input data. Regarding segmentation CTTA (Yang et al., 2023b), the source model is an off-the-shelf pre-trained on the Cityscapes dataset. As for the continual target domains, we utilize the ACDC dataset, which consists of images collected in four unseen visual conditions: Fog, Night, Rain, and Snow. To simulate continual environmental changes in real-life scenarios, we cyclically repeat the same sequence of target domains (Fog→Night→Rain→Snow) multiple times.

**Implementation Details.** In our CTTA experiments, we follow the implementation details specified in previous works (Wang et al., 2022) to ensure consistency and comparability. We adopt ViT-base (Dosovitskiy et al., 2020) and ResNet (He et al., 2016) as the backbone in the classification CTTA. In the case of ViT-base, we resize the input images to 224x224, while maintaining the original image resolution for other backbones. For segmentation CTTA, we adopt the pre-trained Segformer-B5 model (Xie et al., 2021) as the source model. We down-sample the input size from 1920x1080 to 960x540 for target domain data. The optimizer is performed using Adam (Kingma & Ba, 2014) with $(\beta_1, \beta_2) = (0.9, 0.999)$. We set the learning rates to specific values for each task: 1e-4 for CIFAR10C, 5e-7 for ImageNetC, and 3e-4 for ACDC. To initialize our visual domain adapters, we train adapters for several iterations on classification datasets (e.g., ImageNet). We apply a range of image resolution scale factors [0.5, 0.75, 1.0, 1.25, 1.5, 1.75, 2.0] for the augmentation method and construct the teacher model inputs (Wang et al., 2022).

## 4.2 THE EFFECTIVENESS ON CLASSIFICATION CTTA

Table 1: Classification error rate(%) for ImageNet-to-ImageNet-C online CTTA task. Gain(%) represents the percentage of improvement in model accuracy compared with the source method.

| Backbone | Method | REF | Gaussian | shot | impulse | defocus | glass | motion | zoom | snow | frost | fog | brightness | contrast | elastic_trans | pixelate | jpeg | Mean↓ | Gain |
|---|---|---|---|---|---|---|---|---|---|---|---|---|---|---|---|---|---|---|---|
| ResNet50 | Source | (He et al., 2015) | 97.8 | 97.1 | 98.2 | 81.7 | 89.8 | 85.2 | 78 | 83.5 | 77.1 | 75.9 | 41.3 | 94.5 | 82.5 | 79.3 | 68.6 | 82 | 0.0 |
| | TENT | (Wang et al., 2021) | 81.6 | 74.6 | 72.7 | 77.6 | 73.8 | 65.5 | 55.3 | 61.6 | 63 | 51.7 | 38.2 | 72.1 | 50.8 | 47.4 | 53.3 | 62.6 | +19.4 |
| | CoTTA | (Wang et al., 2022) | 84.7 | 82.1 | 80.6 | 81.3 | 79.0 | 68.6 | 57.5 | 60.3 | 60.5 | 48.3 | 36.6 | 66.1 | 47.2 | 41.2 | 46.0 | 62.7 | +19.3 |
| | EcoTTA | (Song et al., 2023) | - | - | - | - | - | - | - | - | - | - | - | - | - | - | - | 63.4 | +18.6 |
| | **Ours** | **Proposed** | **79.3** | **74.7** | **73.1** | **76.9** | **74.5** | **65.0** | **56.4** | **59.8** | **62.6** | **49.6** | **38.2** | **66.8** | **49.6** | **43.1** | **46.2** | **61.2** | **+20.8** |
| ViT-base | Source | (Dosovitskiy et al., 2020) | 53.0 | 51.8 | 52.1 | 68.5 | 78.8 | 58.5 | 63.3 | 49.9 | 54.2 | 57.7 | 26.4 | 91.4 | 57.5 | 38.0 | 36.2 | 55.8 | 0.0 |
| | Pseudo | (Lee, 2013) | 45.2 | 40.4 | 41.6 | 51.3 | 53.9 | 45.6 | 47.7 | 40.4 | 45.7 | 93.8 | 98.5 | 99.9 | 99.9 | 98.9 | 99.6 | 61.2 | -5.4 |
| | TENT | (Wang et al., 2021) | 52.2 | 48.9 | 49.2 | 65.8 | 73 | 54.5 | 58.4 | 44.0 | 47.7 | 50.3 | 23.9 | 72.8 | 55.7 | 34.4 | 33.9 | 51.0 | +4.8 |
| | CoTTA | (Wang et al., 2022) | 52.9 | 51.6 | 51.4 | 68.3 | 78.1 | 57.1 | 62.0 | 48.2 | 52.7 | 55.3 | 25.9 | 90.0 | 56.4 | 36.4 | 35.2 | 54.8 | +1.0 |
| | VDP | (Gan et al., 2023) | 52.7 | 51.6 | 50.1 | 58.1 | 70.2 | 56.1 | 58.1 | 42.1 | 46.1 | 45.8 | 23.6 | 70.4 | 54.9 | 34.5 | 36.1 | 50.0 | +5.8 |
| | **Ours** | **Proposed** | **47.7** | **42.5** | **42.9** | **52.2** | **56.9** | **45.5** | **48.9** | **38.9** | **42.7** | **40.7** | **24.3** | **52.8** | **49.1** | **33.5** | **33.1** | **43.4** | **+12.4** |

**ImageNet-to-ImageNet-C.** Given the source model pre-trained on ImageNet, we conduct CTTA on ImageNet-C, which consists of fifteen corruption types that occur sequentially during the test time. In Table 1, methods that utilize the ViT backbone achieve lower classification errors compared to those using the ResNet50 backbone, demonstrating ViT's superior generalization capability in

Table 2: Average error rate (%) for the standard CIFAR10-to-CIAFAR10C and CIFAR100-to-CIAFAR100C CTTA. All results are evaluated on the ViT-base, and the fine-grained performances are shown in Appendix E.

| Target | Method | Source | Tent | CoTTA | VDP | **Ours** |
|--------|--------|--------|------|-------|-----|----------|
| Cifar10C | Mean↓ | 28.2 | 23.5 | 24.6 | 24.1 | **20.7** |
| | Gain↑ | 0.0 | +4.7 | +3.6 | +4.1 | **+7.5** |
| Cifar100C | Mean↓ | 35.4 | 32.1 | 34.8 | 35.0 | **27.3** |
| | Gain↑ | 0.0 | +3.3 | +0.7 | +0.4 | **+8.1** |

Table 3: Average error rate (%) for the CIFAR10-to-CIFAR10C CTTA task. All results are evaluated on the ViT-Base, which uses the pre-trained encoder parameter of foundation large-scale models (DINOv2 and SAM).

| Backbone | Method | Source | Tent | CoTTA | Ours |
|----------|--------|--------|------|-------|------|
| DINOv2 | Mean↓ | 25.0 | 21.7 | 29.3 | **20.2** |
| | Gain↑ | 0.0 | +3.2 | −4.3 | **+4.8** |
| SAM | Mean↓ | 39.3 | 37.5 | 39.4 | **34.1** |
| | Gain↑ | 0.0 | +1.8 | −0.1 | **+5.2** |

the continually changing environment. For ViT-base, the average classification error is up to 55.8% when we directly test the source model on target domains. Our method can outperform all previous methods, achieving a 12.4% and 6.6% improvement over the source model and previous SOTA method, respectively. Moreover, our method showcases remarkable performance across the majority of corruption types, highlighting its effective mitigation of error accumulation and catastrophic forgetting. In addition, we conduct a 10-round CTTA experiment in Appendix B.1, which repeat 10 rounds of 15 corruption sequences in ImageNet-C. The performance of our method consistently improves over time, demonstrating its enduring robustness in the long-term adaptation process.

To further validate the effectiveness of our method, we conduct experiments on **CIFAR10-to-CIFAR10C** and **CIFAR100-to-CIFAR100C**. As illustrated in Table 2, in CIFAR10C, our approach achieved a 2.8% improvement compared to the previous SOTA model. We extend our evaluation to CIFAR100C, which comprises a larger number of categories in each domain. Our approach surpasses all previous methods, which show the same trend as the above CTTA experiments. Therefore, the results prove that our method mitigates the challenges posed by continual distribution shifts, regardless of the number of categories present in each domain. In addition, we provide supplementary CTTA experiments utilizing convolutional backbones in the Appendix C.5.

## 4.3 THE EFFECTIVENESS ON SEGMENTATION CTTA

Table 4: **Performance comparison for Cityscape-to-ACDC CTTA.** We sequentially repeat the same sequence of target domains three times. Mean is the average score of mIoU.

| | | Time | | | | $t$ ⟶ | | | | | | | | | | | | |
|--------|------|------|------|------|------|------|------|------|------|------|------|------|------|------|------|------|------|------|
| | | Round | | 1 | | | | 2 | | | | 3 | | | | Mean↑ | Gain |
| Method | REF | Fog | Night | Rain | Snow | Mean↑ | Fog | Night | Rain | Snow | Mean↑ | Fog | Night | Rain | Snow | Mean↑ | | |
| Source | (Xie et al., 2021) | 69.1 | 40.3 | 59.7 | 57.8 | 56.7 | 69.1 | 40.3 | 59.7 | 57.8 | 56.7 | 69.1 | 40.3 | 59.7 | 57.8 | 56.7 | 56.7 | / |
| TENT | (Wang et al., 2020) | 69.0 | 40.2 | 60.1 | 57.3 | 56.7 | 68.3 | 39.0 | 60.1 | 56.3 | 55.9 | 67.5 | 37.8 | 59.6 | 55.0 | 55.0 | 55.7 | -1.0 |
| CoTTA | (Wang et al., 2022) | 70.9 | 41.2 | 62.4 | 59.7 | 58.6 | 70.9 | 41.1 | 62.6 | 59.7 | 58.6 | 70.9 | 41.0 | 62.7 | 59.7 | 58.6 | 58.6 | +1.9 |
| DePT | (Gao et al., 2022) | 71.0 | 40.8 | 58.2 | 56.8 | 56.5 | 68.2 | 40.0 | 55.4 | 53.7 | 54.3 | 66.4 | 38.0 | 47.3 | 47.2 | 49.7 | 53.4 | -3.3 |
| VDP | (Gan et al., 2023) | 70.5 | 41.1 | 62.1 | 59.5 | 58.3 | 70.4 | 41.1 | 62.2 | 59.4 | 58.2 | 70.4 | 41.0 | 62.2 | 59.4 | 58.2 | 58.2 | +1.5 |
| **Ours** | **Proposed** | **71.6** | **43.2** | **66.0** | **63.4** | **61.1** | **73.2** | **44.5** | **67.0** | **63.9** | **62.2** | **73.2** | **44.6** | **67.2** | **64.2** | **62.3** | **61.9** | **+5.2** |

**Cityscapes-to-ACDC.** As presented in Table 4, we observed a gradual decrease in the mIoUs of TENT and DePT over time, indicating the occurrence of catastrophic forgetting. In contrast, our method has a continual improvement of average mIoU (61.1→62.2→62.3) when the same sequence of target domains is repeated. Significantly, the proposed method surpasses the previous SOTA CTTA method (Wang et al., 2022) by achieving a 3.3% increase in mIoU. This notable improvement showcases our method's ability to adapt continuously to dynamic target domains in the pixel-level task. The 10 rounds semantic segmentation CTTA experiments are shown in Appendix C.6.

## 4.4 CONTINUAL ADAPTING FOR FOUNDATION MODELS

Foundation models (Bommasani et al., 2021) are trained on large-scale datasets, endowing them with powerful generalization capabilities and the ability to capture representations of common features. However, performing full fine-tuning on the foundation model is time-consuming and economically impractical. Hence, our adaptation method proves valuable by enhancing the continual transfer performance of foundation models. As indicated in Table 3, we introduce foundation models as the pre-trained model and adapt them to continual target domains (CIFAR10C). Our approach achieved a performance improvement of 4.8% on the representative image-level foundation model DINOv2 (Oquab et al., 2023) and 5.2% on pixel-level foundation model SAM (Kirillov et al., 2023).

Table 5: The domain generalization comparisons on ImageNet-C. Results are evaluated on ViT-base. Mean and Gain(%) represent the performance on unseen target domains.

| | Directly test on unseen domains | | | | | Unseen |
|---|---|---|---|---|---|---|
| Method | bri. | contrast | elastic | pixelate | jpeg | Mean↓ |
| Source | 26.4 | 91.4 | 57.5 | 38.0 | 36.2 | 49.9 |
| Tent | 25.8 | 91.9 | 57.0 | 37.2 | 35.7 | 49.5 |
| CoTTA | 25.3 | 88.1 | 55.7 | 36.4 | 34.6 | 48.0 |
| **Ours** | **24.6** | **68.2** | **49.8** | **34.7** | **34.1** | **42.3** |

Table 6: Average error rate (%) for the ImageNet-to-ImageNet-C. $ViDA_h$ and $ViDA_l$ represent the high-rank and low-rank ViDAs. IHKA means inversed HKA strategy.

| | $ViDA_h$ | $ViDA_l$ | HKA | IHKA | Mean↓ |
|---|---|---|---|---|---|
| $Ex_1$ | - | - | - | - | 55.8 |
| $Ex_2$ | ✓ | - | - | - | 50.7 |
| $Ex_3$ | - | ✓ | - | - | 51.2 |
| $Ex_4$ | ✓ | ✓ | - | - | 45.6 |
| $Ex_5$ | ✓ | ✓ | ✓ | - | 43.4 |
| $Ex_6$ | ✓ | ✓ | - | ✓ | 46.3 |

Our method consistently and reliably improves the performance of the foundation model on the continually changing environment. Note that, we only use the pre-trained encoder of SAM and add a classification head, which is fine-tuned on the source domain. Our approach empowers the large-scale model with the capability of continuous transfer learning, without undermining its plasticity. Additional CTTA experiments of foundation models are shown in Appendix C.1 and C.2

### 4.5 DOMAIN GENERALIZATION ON UNSEEN CONTINUAL DOMAINS

To investigate the domain generalization (DG) ability of our method, we follow the leave-one-domain-out rule (Zhou et al., 2021; Li et al., 2017) to leverage 10/15 domains of ImageNet-C as source domains for model training while the rest (5/15 domains) are treated as target domains without any form of adaptation. Specifically, we first use our proposed method to continually adapt the source pre-trained model to 10/15 domains of ImageNet-C without any supervision. Then we directly test on the 5/15 unseen domains. Surprisingly, our method reduces 7.6% on the average error on unseen domains (Table 5), which has a significant improvement over other methods. The promising results demonstrate that our method possesses DG ability by effectively extracting domain-shared knowledge. More DG experiments are provided in the supplementary Appendix C.3.

### 4.6 ABLATION STUDY

**Effectiveness of each component.** We conduct the ablation study on ImageNet-to-ImageNet-C CTTA scenario and evaluate the contribution of each component in our method, including high-rank ViDA ($ViDA_h$), low-rank ViDA ($ViDA_l$), and Homeostatic Knowledge Allotment (HKA) strategy. As shown in Table 6 ($Ex_2$), by introducing the high-rank ViDA, the error decreases by 5.1% compared to $Ex_1$, demonstrating that high-rank features can extract more domain-specific knowledge for adaptation in target domains. As shown in $Ex_3$, low-rank ViDA gains 4.6% improvement compared to $Ex_1$. The result proves that the domain-share knowledge extracted from low-rank feature can also improve the classification ability on continual target domains. $Ex_4$ has a remarkable improvement of 10.2% overall, demonstrating that the two types of ViDA can compensate for each other in the continual adaptation process. $Ex_5$ achieves a 12.4% improvement, demonstrating the effectiveness of the HKA strategy in enhancing the distinct domain representations of each type of ViDA. To further assess the effectiveness of HKA, we perform an additional experiment, denoted as $Ex_6$, by inverting the scale factors within the HKA strategy. Specifically, for samples exhibiting high uncertainty, we reduced $\lambda_h$ while increase $\lambda_l$. This results in a marginal increase of 0.7% error compared to $Ex_4$ and 2.9% error compared to $Ex_5$. The additional ablation studies are shown in Appendix C.4.

## 5 CONCLUSION

In this paper, we propose a homeostatic Visual Domain Adapter (ViDA) to address error accumulation and catastrophic forgetting problems in Continual Test-Time Adaptation (CTTA) tasks. And we investigate that the low-rank ViDA can disregard the impact of dynamic distribution shifts and prioritize the extraction of domain-shared knowledge, and the high-rank ViDA can extract more reliable domain-specific knowledge. Meanwhile, we further propose a Homeostatic Knowledge Allotment (HKA) strategy to dynamically fuse the knowledge from low-rank and high-rank ViDAs, thus enhancing ViDAs' distinct domain representations.

**Acknowledgements.** Shanghang Zhang is supported by the National Science and Technology Major Project of China (No. 2022ZD0117801).

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

## A    APPENDIX

The supplementary materials presented in this paper offer a comprehensive quantitative and qualitative analysis of the proposed method. In Appendix B, we provide additional empirical observations and justifications for our motivation, including specifically designed quantitative analysis, qualitative analysis of the distribution, and justification for distribution divergence. Additionally, we present extra continual adaptation experiments for Foundation Models in Appendices C.1 and C.2, which are conducted on ImageNet-to-ImageNet-C and Cityscape-to-ACDC scenarios. To assess the domain generalization ability of our method, we conducted additional experiments directly testing a varying number of unseen domains in Appendix C.3. The ablation study on middle-layer dimension is described in Appendix C.4. Furthermore, Appendix C.5 presents additional CTTA classification experiments utilizing the convolutional backbone, while Appendix C.6 outlines 10 rounds of semantic segmentation CTTA experiments. We provide an additional qualitative analysis in Appendix D. Moreover, we extend the classification results of our submission to include fine-grained performance in Appendix E, showcasing the error rates across fifteen corruption types.

## B    SUPPLEMENTARY JUSTIFICATIONS FOR MOTIVATION

The study of Continual Test-Time Adaptation (CTTA) poses significant challenges, particularly in addressing error accumulation and catastrophic forgetting (Wang et al., 2022; Gan et al., 2023). Notably, the use of adapters with low-rank and high-rank features have demonstrated promising results in mitigating these challenges in our submission. In this section, we aim to provide comprehensive implementation details regarding the evidence supporting our motivation. Furthermore, we have introduced two new specially designed quantitative experiments in Section B.1. The first one is a 10-round CTTA experiment aimed at investigating the different domain representations of low-rank and high-rank ViDA during the long-term adaptation process. The second experiment explores the performance when all adapters adopt the same structures, such as using two high-rank adapters or two low-rank adapters. This experiment is conducted to validate that low-rank ViDA and high-rank ViDA complement each other in adapting to continually changing environments.

### B.1    SPECIALLY DESIGNED QUANTITATIVE ANALYSIS

To provide stronger evidence for our assumption, we have developed two evaluation approaches for both low-rank and high-rank adapters, which directly reflect their ability to extract domain-shared and domain-specific knowledge on ImageNet-to-ImageNet-C.

First, as shown in Figure 5 (b), we execute a 10 rounds CTTA experiment on ImageNet-to-ImageNet-C. In this comprehensive experiment, we simulate a long-term adaptation scenario by repeating 10 rounds of 15 corruption sequences in the ImageNet-C. Remarkably, the high-rank ViDA achieves competitive results over other methods during the initial 1 to 3 rounds. This result demonstrates the high-rank feature's capacity to efficiently learn target domain-specific knowledge. However, an increment in error rates becomes obvious during the later rounds (rounds 5 to 10). The results validate the potential for encountering catastrophic forgetting when focusing exclusively on domain-specific knowledge. In contrast, the performance of the low-rank ViDA remains consistently robust throughout the continual adaptation process, verifying it concentrates more on extracting task-relevant knowledge and effectively prevents the catastrophic forgetting problem. And our proposed method consistently improves over time, demonstrating its robustness in the long-term adaptation process.

Second, we execute an ImageNet-to-ImageNet-C CTTA experiment using a combination of two high-rank adapters or two low-rank adapters, as shown in Table 7. To ensure fairness, we conducted these experiments without implementing the homeostatic knowledge allotment (HKA) strategy. Notably, the two low-rank adapters (Ex2) demonstrated consistently lower long-term error rates compared to the source model and two high-rank adapters. The above results can be attributed to the fact that the two low-rank ViDAs tend to learn general information and domain-shared knowledge during continual adaptation. However, our method outperforms the two low-rank adapters across 14 out of 15 corruption types. This indicates that solely relying on low-rank adapters without the involvement of high-rank adapters is insufficient to fit target domains and match their data distribution. On the other hand, the performance of the two high-rank adapters initially surpasses our approach (Ex4) in the early stages, covering the first few target domains. Nevertheless, a noticeable performance degradation

Table 7: Classification error rate(%) for ImageNet-to-ImageNet-C online CTTA task. Gain(%) represents the percentage of improvement in model accuracy compared with the source method. 2× means using two same structures of adapters.

| | Method | Gaussian | shot | impulse | defocus | glass | motion | zoom | snow | frost | fog | brightness | contrast | elastic | pixelate | jpeg | Mean↓ |
|---|---|---|---|---|---|---|---|---|---|---|---|---|---|---|---|---|---|
| Ex1 | Source | 53.0 | 51.8 | 52.1 | 68.5 | 78.8 | 58.5 | 63.3 | 49.9 | 54.2 | 57.7 | 26.4 | 91.4 | 57.5 | 38.0 | 36.2 | 55.8 |
| Ex2 | 2×Low-rank | 51.2 | 48.3 | 47.8 | 56.9 | 66.5 | 49.3 | 54.4 | 42.1 | 47.0 | 45.2 | 23.2 | 65.6 | 52.0 | **33.4** | 33.5 | 47.7 |
| Ex3 | 2×High-rank | **50.1** | 47.9 | **45.3** | **54.8** | 66.7 | 51.4 | 56.1 | 44.0 | 49.2 | 48.3 | 25.7 | 69.7 | 56.3 | 34.6 | 33.7 | 48.9 |
| Ex4 | Ours | 50.3 | **45.9** | 45.5 | 55.1 | **62.3** | **46.6** | **51.7** | **39.7** | **44.0** | **42.2** | **23.0** | **62.4** | **50.1** | 33.4 | **32.5** | **45.6** |

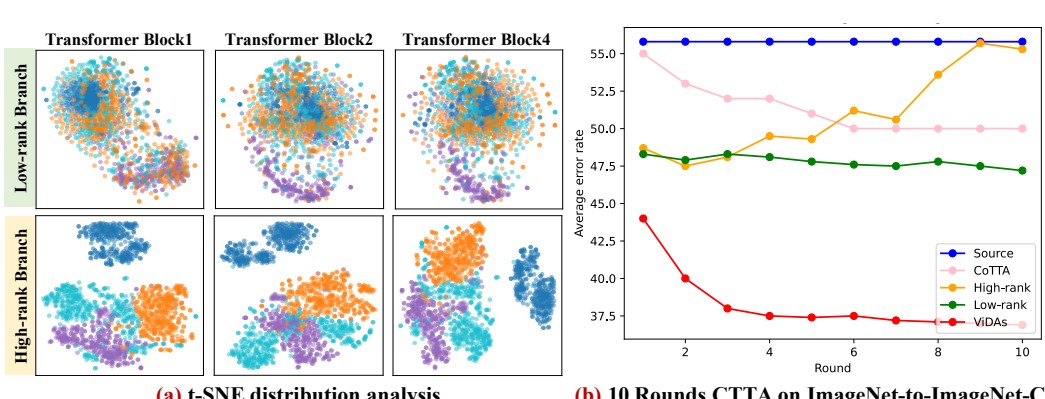

(a) t-SNE distribution analysis    (b) 10 Rounds CTTA on ImageNet-to-ImageNet-C

Figure 5: (a) We conduct more t-SNE results for the low-rank adapter and high-rank adapter on the ACDC dataset. The first to third columns illustrate the feature distributions of transformer blocks 1, 2, and 4, respectively. (b) The 10 rounds CTTA experiment on ImageNet-to-ImageNet-C, repeating 10 rounds of 15 corruption sequences.

becomes apparent in later target domains. This observation underscores a crucial finding: while increasing the number of high-rank ViDAs might enhance domain-specific knowledge acquisition during the initial phases of CTTA, it simultaneously exacerbates catastrophic forgetting throughout the entire adaptation process. In contrast, the fusion of both low-rank and high-rank ViDAs (Ex4) yields the most substantial improvement when compared to other configurations. Our collaborative approach leverages the distinct domain representations of these adapters to compensate for each other's advantages and achieve a more robust and effective continual adaptation.

## B.2 ADDITIONAL DISTRIBUTION QUALITATIVE ANALYSIS

We employed t-distributed stochastic neighbor embedding (t-SNE) (Van der Maaten & Hinton, 2008) to visualize the distribution of adapters across four continual target domains. This visualization was specifically conducted in the context of the Cityscapes-to-ACDC experiment, representing a scenario with continually changing real-world environments.In our submission, we perform t-SNE analysis on the outputs of the third transformer block in the Segformer-B5 model (Xie et al., 2021). The objective was to qualitatively compare the feature distributions of ViDAs with different dimension features. Furthermore, our findings revealed that the qualitative results obtained from different layers (i.e., transformer block 1, 2, and 4) of the Segformer-B5 model exhibited similar distribution representations. As illustrated in Figure 5 (a), there is a noticeable distribution gap due to the significant domain shift between the night domain and other domains. Interestingly, the low-rank ViDA effectively reduces the distribution distance across different target domains, indicating its focus on extracting task-relevant knowledge. On the other hand, the high-rank ViDA exhibits notable distribution discrepancies among the various target domains, indicating its focus on extracting domain-specific knowledge.

### B.3 DISTRIBUTION DISTANCE

To provide clearer evidence for our assumption, we directly calculate the distribution distance to represent different domain representation of adapters. We adopt the domain distance definition proposed by Ben-David (Ben-David et al., 2006; 2010) and build upon previous domain transfer research (Ganin et al., 2016) by employing the $\mathcal{H}$-$divergence$ metric to further evaluate the domain representations of adapters across different target domains. $\mathcal{H}$-$divergence$ between $D_S$ and $D_{T_i}$ can be calculated as

$$d_{\mathcal{H}}(D_S, D_{T_i}) = 2 \sup_{\mathcal{D} \sim \mathcal{H}} | \Pr_{x \sim D_S} [\mathcal{D}(x) = 1] - \Pr_{x \sim D_{T_i}} [\mathcal{D}(x) = 1]| \tag{8}$$

, where $\mathcal{H}$ denotes hypothetical space and $\mathcal{D}$ denotes discriminator. Similar to (Ruder & Plank, 2017; Allaway et al., 2021), we adopt the $Jensen\text{-}Shannon\ (JS)\ divergence$ between two adjacent domains as an approximation of $\mathcal{H}$-divergence because it has been shown to successfully distinguish domains. If the inter-domain divergence is relatively small, it can be demonstrated that the feature representation is consistent and less influenced by cross-domain shifts (Ganin et al., 2016).

$$JS(P_{D_S}||P_{D_{T_i}}) = \frac{1}{2} KL(P_{D_S}||\frac{P_{D_S} + P_{D_{T_i}}}{2}) + \frac{1}{2} KL(P_{D_{T_i}}||\frac{P_{D_S} + P_{D_{T_i}}}{2}) \tag{9}$$

Where $Kullback\text{-}Leibler\ (KL)\ divergence$ between two domain is

$$KL(P_1||P_2) = \sum_{i=0}^{n} P_1(x_i) log(\frac{P_1(x_i)}{P_2(x_i)}) \tag{10}$$

Where $P$ denotes probability distribution of model output features. We split the output feature space into mutually disjoint intervals $x_i$. $n$ range from 0 to 1000. To investigate the effectiveness of adapters in adapting to continual target domains, we compare the $JS$ values obtained by using the source model alone, injecting low-rank adapter, injecting high-rank adapter, and combining low-high adapters, as illustrated in Figure 3(a) of our submission. The low-rank adapter exhibits notably lower divergence values compared to the others, demonstrating robust task-relevant feature representation in various cross-domain phases. For high-rank adapter, we use normalized intra-class divergence to further verify the domain representations of high-rank adapters in CIFAR10C, which is inspired by intra-cluster dissimilarity proposed by $k$-means (MacQueen, 1967). We first calculate the Euclidean distance clustering center for each category:

$$\mu = \frac{1}{|C|} \sum_{e_i \sim C} e_i \tag{11}$$

, where $e_i$ stands for output feature in class $C$. Then following (MacQueen, 1967), we introduce normalized intra-class divergence $E$ by

$$E = \phi(\frac{1}{|C|} \sum_{e_i \sim C} ||e_i - \mu||_2^2) \tag{12}$$

$\phi(\cdot)$ denotes for normlization function. In a given domain, if the intra-class divergence for each category is smaller, it demonstrates that the model has a better understanding of the current distribution (Li et al., 2020). As illustrated in Figure 3(b) of the submission, the high-rank adapter is found to drive down divergence within almost all domains and can better extract domain-specific knowledge in target domains.

## C ADDITIONAL EXPERIMENT

### C.1 ADDITIONAL CLASSIFICATION CTTA EXPERIMENTS FOR FOUNDATION MODELS

To demonstrate the effectiveness of our proposed method in enhancing the continual adaptation ability of foundation models such as DINOv2 (Oquab et al., 2023) and SAM (Kirillov et al., 2023), we conduct additional experiments on a more extensive dataset, namely ImagNet-to-ImageNet-C. Our approach involve loading the weight parameters of the foundation model and fine-tuning it

Table 8: Average error rate (%) for the ImageNet-to-ImageNet-C CTTA task. All results are evaluated on the ViT-Base, which uses the pre-trained encoder parameter of DINOv2 and SAM.

| Backbone | Method | REF | Gaussian | shot | impulse | defocus | glass | motion | zoom | snow | frost | fog | brightness | contrast | elastic_trans | pixelate | jpeg | Mean↓ | Gain |
|---|---|---|---|---|---|---|---|---|---|---|---|---|---|---|---|---|---|---|---|
| DINOv2 | Source | | 52.3 | 50.5 | 51.2 | 57.3 | 83.8 | 60.1 | 62.6 | 47.1 | 56.9 | 58.1 | 22.5 | 88.4 | 60.3 | 32.4 | 35.0 | 54.6 | 0.0 |
| | Tent (Wang et al., 2021) | ICLR2021 | 51.7 | 43.6 | 50.4 | 56.2 | 74.1 | 51.7 | 67.2 | 46.9 | 53.2 | 50.1 | 25.2 | 69.6 | 58.0 | 29.5 | 39.4 | 51.1 | +3.5 |
| | CoTTA (Wang et al., 2022) | CVPR2022 | 51.4 | 62.1 | 50.4 | 78.3 | 75.2 | 62.8 | 60.3 | 48.4 | 59.0 | 58.8 | 31.6 | 90.7 | 49.2 | 39.1 | 36.5 | 56.9 | -2.3 |
| | **Ours** | Proposed | **49.0** | 49.8 | 50.7 | 61.4 | **60.2** | 49.7 | 42.6 | 47.1 | 51.9 | 45.3 | 27.1 | 49.7 | 47.4 | 32.0 | **29.4** | **46.2** | **+8.4** |
| SAM | Source | | 67.9 | 62.1 | 51.6 | 69.7 | 92.6 | 65.4 | 59.8 | 53.9 | 61.2 | 64.1 | 39.0 | 91.6 | 60.1 | 47.3 | 67.0 | 63.6 | 0.0 |
| | Tent (Wang et al., 2021) | ICLR2021 | 67.2 | 59.1 | 48.8 | 56.2 | 72.5 | 59.4 | 61.0 | 49.1 | 57.9 | 63.7 | 33.8 | 77.0 | 51.4 | 39.5 | 55.2 | 55.5 | +8.1 |
| | CoTTA (Wang et al., 2022) | CVPR2022 | 68.1 | 64.5 | 50.4 | 67.1 | 80.1 | 68.9 | 67.0 | 63.1 | 69.5 | 61.4 | 40.6 | 88.2 | 58.3 | 43.5 | 68.4 | 63.9 | -0.3 |
| | **Ours** | Proposed | **59.9** | **55.7** | **40.2** | 84.3 | **49.6** | 59.7 | **59.0** | 47.8 | 48.3 | 57.4 | 26.6 | 71.8 | 42.9 | 41.7 | 50.3 | **53.0** | **+10.6** |

on ImagNet, thus constructing our source model. It is important to note that we solely utilize the pre-trained encoder of SAM and incorporated a classification head, which is fine-tuned on the source domain. Subsequently, we adapt the source model to continual target domains (ImageNet-C) comprising fifteen corruption types. The results, as depicted in Table 8, demonstrate that our approach achieved a significant performance improvement of 8.4% on the representative image-level foundation model DINOv2 and 10.6% on the pixel-level foundation model SAM. These outcomes underscore the effectiveness of our method for large-scale models, consistently and reliably improving performance across target domains. Combining Table 1-3 from the submission, we were surprised to discover a significant decrease in model performance for the classification CTTA task when using the pre-trained encoder parameters of SAM. As SAM is a pixel-level foundation model, we then attempted to investigate the effectiveness of SAM's pretrained parameters in the segmentation CTTA task.

## C.2 ADDITIONAL SEGMENTATION CTTA EXPERIMENTS FOR FOUNDATION MODELS

As shown in Table 9, we conducted segmentation CTTA using SAM's pre-trained parameters on the Cityscapes-to-ACDC scenario. However, it's worth noting that the Segformer model (Xie et al., 2021), which we employed in our main experiments, does not incorporate positional encoding. Therefore, we adopted the SETR model (Zheng et al., 2021) as our new baseline for loading SAM's pre-trained parameters. As shown in the table, our approach with SAM's pre-trained parameters outperforms others on the ACDC target domains. This aligns with the assumption that SAM, being a pixel-level foundational model, excels in capturing fine-grained feature representations in dense CTTA tasks.

Table 9: Performance comparison for Cityscape-to-ACDC CTTA. All results are evaluated on the SETR, which uses the pre-trained parameter of source model or SAM.

| Method | Pre-trained | Fog | Night | Rain | Snow | Mean mIoU |
|---|---|---|---|---|---|---|
| Source (Xie et al., 2021) | Source model | 72.6 | 43.1 | 63.0 | 64.3 | 60.8 |
| Source (Xie et al., 2021) | SAM (Kirillov et al., 2023) | 74.8 | 44.1 | 66.7 | 66.6 | 63.0 |
| CoTTA (Wang et al., 2022) | SAM (Kirillov et al., 2023) | 75.4 | 45.9 | 67.3 | 68.7 | 64.3 |
| **Ours** | SAM (Kirillov et al., 2023) | **76.5** | **47.2** | **68.1** | **70.7** | **65.6** |

## C.3 DOMAIN GENERALIZATION ON A DIFFERENT NUMBER OF UNSEEN TARGET DOMAINS

Similar to our previous submission, we follow the leave-one-domain-out principle (Zhou et al., 2021; Li et al., 2017), where we utilize a subset of ImageNet-C domains as new source domains for model training, while leaving the remaining domains as target domains without any adaptation. However, in contrast to previous domain generalization experiments, we adopt an unsupervised continual test-time adaptation (CTTA) approach for training the model on these unlabeled source domains. We solely utilize the ImageNet pre-trained parameters as the initial weights of the model. In the supplementary material, we utilize 5 out of 15 and 7 out of 15 domains from ImageNet-C as the source domains, leaving the remaining 10 out of 15 and 8 out of 15 domains as unseen target domains. Surprisingly, the results presented in Table 10 and 11 demonstrate that our method achieves a reduction of 9.6% and 9.1% in the average error on these unseen domains, respectively. These promising outcomes validate the DG ability of our method, as it effectively extracts domain-shared knowledge and provides a new perspective for enhancing DG performance within an unsupervised paradigm.

Table 10: The domain generalization experiments on ImageNet-C, where the source model was continually adapted on the first 5 domains and directly tested on 10 unseen domains. The evaluation of the results was conducted using ViT-base.

| Method | Directly test on 10 unseen domains | | | | | | | | | | Unseen |
| | motion | zoom | snow | frost | fog | brightness | contrast | elastic_trans | pixelate | jpeg | Mean↓ |
|---|---|---|---|---|---|---|---|---|---|---|---|
| Source | 58.5 | 63.3 | 49.9 | 54.2 | 57.7 | 26.4 | 91.4 | 57.5 | 38.0 | 36.2 | 53.3 |
| Tent (Wang et al., 2021) | 56.0 | 61.3 | 45.7 | 49.6 | 56.6 | 24.8 | 94.0 | 55.6 | 37.1 | 35.1 | 51.6 |
| CoTTA (Wang et al., 2022) | 57.3 | 62.1 | 49.1 | 52.0 | 57.1 | 26.4 | 91.9 | 57.1 | 37.6 | 35.3 | 52.6 |
| **Ours** | **46.4** | **52.7** | **39.8** | **43.7** | **42.2** | **23.5** | **71.5** | **49.6** | **33.9** | **33.3** | **43.7** |

Table 11: The domain generalization experiments on ImageNet-C, where the source model was continually adapted on the first 7 domains and directly tested on 8 unseen domains. The evaluation of the results was conducted using ViT-base.

| Method | Directly test on 8 unseen domains | | | | | | | | Unseen |
| | snow | frost | fog | brightness | contrast | elastic_trans | pixelate | jpeg | Mean↓ |
|---|---|---|---|---|---|---|---|---|---|
| Source | 49.9 | 54.2 | 57.7 | 26.4 | 91.4 | 57.5 | 38.0 | 36.2 | 51.4 |
| Tent (Wang et al., 2021) | 44.3 | 48.8 | 51.8 | 24.9 | 83.7 | 55.2 | 35.4 | 34.7 | 47.4 |
| CoTTA (Wang et al., 2022) | 48.8 | 52.2 | 56.7 | 26.1 | 91.1 | 57.0 | 37.3 | 35.3 | 50.6 |
| **Ours** | **39.6** | **43.7** | **41.7** | **23.7** | **63.7** | **51.7** | **33.3** | **33.6** | **42.3** |

## C.4 ADDITIONAL ABLATION STUDY

**How does the middle-layer dimension influence the performance?**

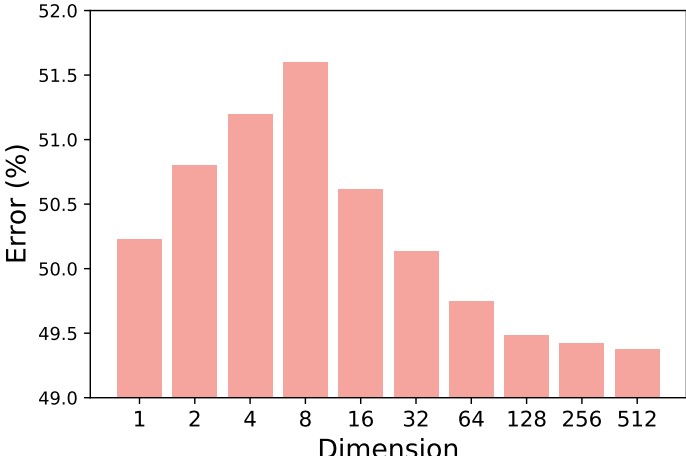

Figure 6: The middle-layer dimension influence of the performance

According to Figure 6, we observe that as the dimension decreases, the error rate concurrently drops. This trend suggests that lower-dimension middle layer more effectively extract the domain-shared knowledge, leading to an improved model performance. However, an opposite trend emerges when dimension surpasses 16, with performance enhancements accompanying increased dimension. This correlation implies that middle layers with a higher dimension excel in extracting domain-specific knowledge. And we find that when the dimension is larger than 128, the performance improvement

is limited but brings a larger number of parameters. Therefore, we set the dimension of the high-dimension middle layer to 128 in our study.

**How do different adapter initialization methods impact ViDA performance?**

Pre-training the low-rank and high-rank ViDAs using source data is an unnecessary step and does not compromise the effectiveness of our approach. ViDAs can demonstrate comparable CTTA performance when they have a relatively stable initial parameter. As illustrated in the Table 12, we conduct an additional experiment on the Cityscape-to-ACDC scenario. ViDAs with random initial parameters and ViDAs with parameters pre-trained on ImageNet achieved 60.5 and 61.4 mIoU in target domains, respectively, exhibiting notable improvements compared to previous methods.

Table 12: The ablation study examines adapter initialization methods on the Cityscape-to-ACDC CTTA scenario.

| | Adapter pre-train | Fog | Night | Rain | Snow | Mean (IoU) |
|---|---|---|---|---|---|---|
| Source | - | 69.1 | 40.3 | 59.7 | 57.8 | 56.7 |
| CoTTA | - | 70.9 | 41.2 | 62.4 | 59.7 | 58.6 |
| Ours | Source | 71.6 | 43.2 | 66.0 | 63.4 | 61.1 |
| Ours | Random initial | 71.6 | 43.6 | 64.9 | 61.9 | 60.5 |
| Ours | ImageNet | 71.6 | 44.3 | 66.0 | 63.5 | 61.4 |

## C.5 EXPERIMENTS ON CLASSIFICATION CTTA WITH CONVOLUTIONAL BACKBONES

Table 13: Classification error rate(%) for standard CIFAR10-to-CIAFAR10C online CTTA task. Results are evaluated on WideResNet-28. Mean is the average value of the error rate. Gain(%) represents the percentage of improvement in model accuracy compared with the source method.

| Method | REF | Conference | Mean↓ | Gain |
|---|---|---|---|---|
| Source | (Zagoruyko & Komodakis, 2016) | BMVC2016 | 43.5 | 0.0 |
| BN Stats Adapt | (Schneider et al., 2020) | NeurIPS2020 | 20.4 | +23.1 |
| TENT | (Wang et al., 2021) | ICLR2021 | 20.7 | +22.8 |
| CoTTA | (Wang et al., 2022) | CVPR2022 | 16.2 | +27.3 |
| RoTTA | (Yuan et al., 2023) | CVPR2023 | 17.5 | +26.0 |
| NOTE | (Gong et al., 2022) | NeurIPS2022 | 20.2 | +23.3 |
| EcoTTA | (Song et al., 2023) | ICCV2023 | 16.8 | +26.7 |
| SATA | (Chakrabarty et al., 2023) | 2023.4.20 | 16.1 | +27.4 |
| **Ours** | Proposed | 2023.5.18 | **15.8** | **+27.7** |

**CIFAR10-to-CIFAR10C standard task.** In contrast to the experiments conducted in our submission, we introduce a change in the backbone of the classification model to WideResNet-28, which is consistent with previous works (Wang et al., 2022). Specifically, we modify the up-projection layer and down-projection layer to utilize $1 \times 1$ convolutions, while the adapters are placed alongside the original $3 \times 3$ convolutions. For ViDA, we maintain a low-rank dimension of 1 and a high-rank dimension of 128. As depicted in Table 13, our method achieves a 27.7% improvement over the source model. These findings demonstrate that our method successfully address error accumulation and catastrophic forgetting problem, regardless of the network backbone employed.

## C.6 ADDITIONAL EXPERIMENTS ON SEGMENTATION CTTA

We further present the segmentation CTTA experiment with 10 rounds on Table 14. Notably, it demonstrates a consistent enhancement in mean mIoU during the initial rounds (rounds 1-3) while maintaining stable performance in subsequent rounds (rounds 4-10). After averaging over 10 rounds , our method achieved a 3.0% mIoU improvement compared to the previous SOTA method. As shown in Table 14 (CoTTA*), we adjust the hyperparameters of the CoTTA method by raising the learning rate to 3e-4, which aligns with our implementation details. The impact of this adjustment is evident in the initial three rounds of segmentation, where performance notably improves. However, as we progress to subsequent CTTA rounds, we observe a noticeable decline in segmentation accuracy and encounter the problem of catastrophic forgetting.

Table 14: **10 rounds segmentation CTTA on Cityscape-to-ACDC.** We sequentially repeat the same sequence of target domains 10 times. Mean is the average score of mIoU.

| Round | 1 | | | | | 2 | | | | | 3 | | | | | 4 | | | | | 5 | | | | | Mean |
|---|---|---|---|---|---|---|---|---|---|---|---|---|---|---|---|---|---|---|---|---|---|---|---|---|---|---|
| Method | Fog | Night | Rain | Snow | Mean | Fog | Night | Rain | Snow | Mean | Fog | Night | Rain | Snow | Mean | Fog | Night | Rain | Snow | Mean | Fog | Night | Rain | Snow | Mean | |
| Source | 69.1 | 40.3 | 59.7 | 57.8 | 56.7 | 69.1 | 40.3 | 59.7 | 57.8 | 56.7 | 69.1 | 40.3 | 59.7 | 57.8 | 56.7 | 56.7 | 40.3 | 59.7 | 57.8 | 56.7 | 56.7 | 40.3 | 59.7 | 57.8 | 56.7 | cont. |
| CoTTA | 70.9 | 41.2 | 62.4 | 59.7 | 58.6 | 70.9 | 41.1 | 62.6 | 59.7 | 58.6 | 70.9 | 41.0 | 62.7 | 59.7 | 58.6 | 70.9 | 41.0 | 62.7 | 59.7 | 58.6 | 70.9 | 41.0 | 62.8 | 59.7 | 58.6 | cont. |
| CoTTA* | 71.9 | 45.0 | 67.1 | 63.1 | 61.8 | 71.9 | 43.6 | 65.6 | 61.8 | 60.7 | 69.6 | 39.7 | 63.5 | 60.4 | 58.3 | 68.3 | 39.6 | 61.8 | 59.4 | 57.3 | 67.8 | 38.9 | 62.1 | 59.7 | 57.1 | cont. |
| Ours | 71.6 | 43.2 | 66.0 | 63.4 | **61.1** | 73.2 | 44.5 | 67.0 | 63.9 | **62.2** | 73.2 | 44.6 | 67.2 | 64.2 | **62.3** | 70.9 | 44.0 | 66.0 | 63.2 | **61.0** | 72.0 | 43.7 | 66.3 | 63.1 | **61.3** | cont. |
| Round | 6 | | | | | 7 | | | | | 8 | | | | | 9 | | | | | 10 | | | | | Mean |
| Method | Fog | Night | Rain | Snow | Mean | Fog | Night | Rain | Snow | Mean | Fog | Night | Rain | Snow | Mean | Fog | Night | Rain | Snow | Mean | Fog | Night | Rain | Snow | Mean | |
| Source | 69.1 | 40.3 | 59.7 | 57.8 | 56.7 | 69.1 | 40.3 | 59.7 | 57.8 | 56.7 | 69.1 | 40.3 | 59.7 | 57.8 | 56.7 | 56.7 | 40.3 | 59.7 | 57.8 | 56.7 | 56.7 | 40.3 | 59.7 | 57.8 | 56.7 | 56.7 |
| CoTTA | 70.9 | 41.0 | 62.8 | 59.7 | 58.6 | 70.9 | 41.1 | 62.6 | 59.7 | 58.6 | 70.9 | 41.1 | 62.6 | 59.7 | 58.6 | 70.8 | 41.1 | 62.6 | 59.7 | 58.6 | 70.8 | 41.1 | 62.6 | 59.7 | 58.6 | 58.6 |
| CoTTA* | 67.7 | 39.8 | 62.7 | 59.7 | 57.5 | 67.3 | 39.7 | 63.2 | 59.6 | 57.7 | 67.6 | 40.1 | 63.2 | 58.0 | 57.2 | 65.0 | 38.8 | 60.7 | 58.5 | 55.8 | 66.9 | 38.9 | 62.7 | 58.7 | 56.8 | 58.0 |
| Ours | 72.2 | 44.0 | 66.6 | 62.9 | **61.4** | 72.3 | 44.8 | 66.5 | 62.9 | **61.6** | 72.1 | 45.1 | 66.2 | 62.9 | **61.5** | 71.9 | 45.3 | 66.3 | 62.9 | **61.5** | 72.2 | 45.2 | 66.5 | 62.9 | **61.6** | **61.6** |

# D    ADDITIONAL QUALITATIVE ANALYSIS

To further validate the effectiveness of our proposed method, we present additional qualitative comparisons on the Cityscapes-to-ACDC CTTA scenario. Initially, we pre-train the Segformer-B5 model (Xie et al., 2021) on the source domain and subsequently adapt it to four target domains in ACDC. In order to assess the performance of our approach, we conduct a qualitative comparison with two leading methods, namely CoTTA (Wang et al., 2022) and VDP (Gan et al., 2023). The visualizations of the segmentation outputs, obtained through the CTTA process, are depicted in Figure 7. Our method exhibits better segmentation map compared to CoTTA and VDP across all four target domains, as it effectively distinguishes the sidewalk from the road (shown in white box). This demonstrates the capability of our method to achieve more accurate segmentation results while mitigating the impact of dynamic domain shifts. Moreover, in the other categories, our method's segmentation maps closely resemble the Ground Truth, leading to a visual enhancements. Lastly, we have included a video visualization in the supplementary material that showcases a comprehensive comparison of segmentation performance. This video provides a dynamic and visual representation of the results obtained from our experiments.

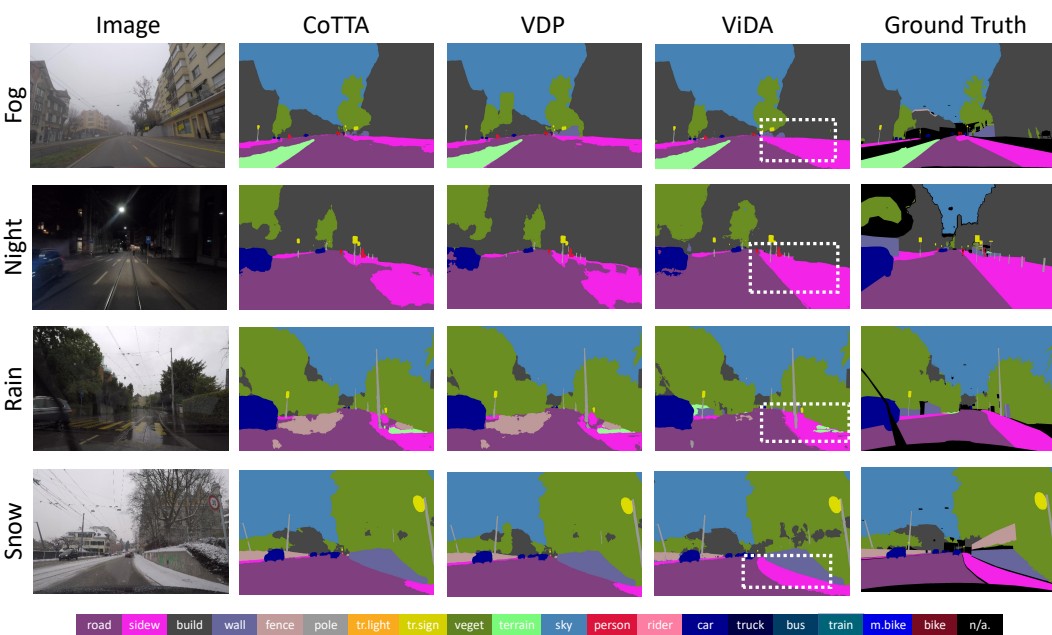

Figure 7: Qualitative comparison of our method with previous SOTA methods on the ACDC dataset. Our method could better segment different pixel-wise classes such as shown in the white box.

Table 15: A fine-grained Classification error rate(%) for standard CIFAR10-to-CIAFAR10C online CTTA task. Results are evaluated on ViT-base.

| Method | gaussion | shot | impulse | defocus | glass | motion | zoom | snow | frost | fog | bri. | contrast | elastic_trans | pixelate | jpeg | Mean↓ | Gain |
|---|---|---|---|---|---|---|---|---|---|---|---|---|---|---|---|---|---|
| Source | 60.1 | 53.2 | 38.3 | 19.9 | 35.5 | 22.6 | 18.6 | 12.1 | 12.7 | 22.8 | 5.3 | 49.7 | 23.6 | 24.7 | 23.1 | 28.2 | 0.0 |
| Pseudo-label (Lee, 2013) | 59.8 | 52.5 | 37.2 | 19.8 | 35.2 | 21.8 | 17.6 | 11.6 | 12.3 | 20.7 | 5.0 | 41.7 | 21.5 | 25.2 | 22.1 | 26.9 | +1.3 |
| TENT-continual (Wang et al., 2021) | 57.7 | 56.3 | 29.4 | 16.2 | 35.3 | 16.2 | 12.4 | 11.0 | 11.6 | 14.9 | 4.7 | 22.5 | 15.9 | 29.1 | 19.5 | 23.5 | +4.7 |
| CoTTA (Wang et al., 2022) | 58.7 | 51.3 | 33.0 | 20.1 | 34.8 | 20 | 15.2 | 11.1 | 11.3 | 18.5 | 4.0 | 34.7 | 18.8 | 19.0 | 17.9 | 24.6 | +3.6 |
| VDP(Gan et al., 2023) | 57.5 | 49.5 | 31.7 | 21.3 | 35.1 | 19.6 | 15.1 | 10.8 | 10.3 | 18.1 | 4 | 27.5 | 18.4 | 22.5 | 19.9 | 24.1 | +4.1 |
| **Ours (proposed)** | **52.9** | **47.9** | **19.4** | **11.4** | **31.3** | **13.3** | **7.6** | **7.6** | **9.9** | **12.5** | **3.8** | **26.3** | **14.4** | **33.9** | **18.2** | **20.7** | **+7.5** |

Table 16: A fine-grained Classification error rate(%) for standard CIFAR100-to-CIAFAR100C online CTTA task. Results are evaluated on ViT-base.

| Method | gaussion | shot | impulse | defocus | glass | motion | zoom | snow | frost | fog | bri. | contrast | elastic_trans | pixelate | jpeg | Mean↓ | Gain |
|---|---|---|---|---|---|---|---|---|---|---|---|---|---|---|---|---|---|
| Source | 55.0 | 51.5 | 26.9 | 24.0 | 60.5 | 29.0 | 21.4 | 21.1 | 25.0 | 35.2 | 11.8 | 34.8 | 43.2 | 56.0 | 35.9 | 35.4 | 0.0 |
| Pseudo-label (Lee, 2013) | 53.8 | 48.9 | 25.4 | 23.0 | 58.7 | 27.3 | 19.6 | 20.6 | 23.4 | 31.3 | 11.8 | 28.4 | 39.6 | 52.3 | 33.9 | 33.2 | +2.2 |
| TENT-continual (Wang et al., 2021) | 53.0 | 47.0 | 24.6 | 22.3 | 58.5 | 26.5 | 19.0 | 21.0 | 23.0 | 30.1 | 11.8 | 25.2 | 39.0 | 47.1 | 33.3 | 32.1 | +3.3 |
| CoTTA (Wang et al., 2022) | 55.0 | 51.3 | 25.8 | 24.1 | 59.2 | 28.9 | 21.4 | 21.0 | 24.7 | 34.9 | 11.7 | 31.7 | 40.4 | 55.7 | 35.6 | 34.8 | +0.6 |
| VDP (Gan et al., 2023) | 54.8 | 51.2 | 25.6 | 24.2 | 59.1 | 28.8 | 21.2 | 20.5 | 23.3 | 33.8 | 7.5 | 11.7 | 32.0 | 51.7 | 35.2 | 32.0 | +3.4 |
| **Ours (proposed)** | **50.1** | **40.7** | **22.0** | **21.2** | **45.2** | **21.6** | **16.5** | **17.9** | **16.6** | **25.6** | **11.5** | **29.0** | **29.6** | **34.7** | **27.1** | **27.3** | **+8.1** |

# E  FINE-GRAINED PERFORMANCE

In this section, we expand upon the classification results presented in our submission by providing a details of fine-grained performance. We assess the error rates across fifteen corruption types to gain deeper insights. To be specific, we augment the information provided in Table 2 of our submission with the additional details presented in Table 15 and 16. These tables offer a comprehensive view of the performance of our approach in addressing the CIFAR-10-to-CIFAR-10C and CIFAR-100-to-CIFAR-100C CTTA scenarios, respectively.

