# OpenReview forum: "ViDA: Homeostatic Visual Domain Adapter for Continual Test Time Adaptation"
_ICLR.cc/2024/Conference — ICLR 2024 poster_

### Official Review · Reviewer_Ut2z · 2023-10-31

**Soundness:** 3 good
**Presentation:** 3 good
**Contribution:** 3 good
**Rating:** 8
**Confidence:** 4

**Summary:**

The authors have proposed a method for continual test time adaptation and are improving previous methods significantly. They do so by incorporating a low-rank and a high-rank embedding scheme where high-rank scheme is shown to be sensitive towards domain specific features whereas low-rank scheme is shown to be sensitive towards domain-invariant features. Additionally, to weigh these features during adaptation, they introduced a homeostatic knowledge allotment strategy where they exploit the probabilistic nature of multiple forward passes to get the best set of parameters. Overall, the authors are achieving reasonably well results.

**Strengths:**

Continual Test Time Adaptation is a relatively new and challenging problem in the vision community. The authors have incorporated a paradigm that does just that.

Originality
- This particular challenge is still at its infancy and the proposed ideas in this paper can help mature the challenge. In terms of originality, the authors have incorporated similar to [1] with their domain specific and domain invariant learning scheme; however, these proposed ideas have shown to contribute significantly, and they have shown it in the results. Additionally, their homeostatic knowledge allotment scheme is also quite useful in terms of what features should weigh more than the other. Therefore, I believe the work is original and is contributing to the field.

Quality
- Since the challenge has a very big  real world applicability, and the approach has shown to perform very well in this regard.
- Additionally, the authors have shown extensive experiments on various standard datasets where they outperform other SOTA approaches including CoTTA [2] and VDP [1].
- The authors have shown extensive ablation study and motivation to use the different components of the approach in the paper.

Clarity
- The paper is well written and easy to follow.
- The authors have clarified their statements with equations and experiments.

Significance
- As mentioned previously, the CTTA has a lot of real-world applicability in IID and non-IID scenarios, and recently, works have been tackling both of them in different scenarios. This work has shown perform boost in this regard, and the authors have solifdified their claims with extensive experiments.
- Additionally, as the paper mentions, it can be integrated with any networks very easily further boosting the applicability of the method.
- This work has potential to contribute significantly to the community.

References:
1. Gan, Y., Bai, Y., Lou, Y., Ma, X., Zhang, R., Shi, N. and Luo, L., 2023, June. Decorate the newcomers: Visual domain prompt for continual test time adaptation. In Proceedings of the AAAI Conference on Artificial Intelligence (Vol. 37, No. 6, pp. 7595-7603).
2. Wang, Q., Fink, O., Van Gool, L. and Dai, D., 2022. Continual test-time domain adaptation. In Proceedings of the IEEE/CVF Conference on Computer Vision and Pattern Recognition (pp. 7201-7211).

**Weaknesses:**

Concerns
- The authors mentioned that there is not any non-linear layer in ViDA. It is not clear what they mean by that. There is no activation? How are they implementing this network? The authors do mention that there is a direct linear projection of ViDA; however, I would like to understand the relationship of the layers within the model.
- How did the authors decide parameter 𝜭 in equation 5? Was it decided empirically?

**Questions:**

I recommend clarifying some confusions from the weakness section about the non-linearity of the model.
- In what sense the is not any non-linear layer in ViDA? Is there no activation? How are you implementing this network? Is there a direct linear projection of ViDA? It would be important to understand the relationship of the layers within the model.
- How was the parameter 𝜭 in equation 5 set?
- Additionally, I suggest incorporating details such as the ablation study on homeostatic knowledge allotment from the appendix to the main paper.

---

> ### Author Response · Authors · 2023-11-17
> **Rebuttal response to Reviewer Ut2z (Q1)**
>
> **Q1. The implementation of ViDA**
>
> Thank you for your insightful comments. As illustrated in Figure 2 of the submission, there are three sub-branches. The single linear layer in the middle branch is derived from the original network, while the right and left branches with bottleneck structures, representing the high-rank ViDA and low-rank ViDA, respectively.
> In both ViDA branches, two linear layers (fully connected layers) handle the transformation of feature dimensionality, **and there are no activation layers, such as ReLU and Leaky ReLU**.  For example, in the high-rank branch, we first employ the up-projection layer with parameters $W^h_{up} \in R^{d \times d_{h}}$, which elevates the feature dimensionality from $B \times  N \times  d$ to $B \times  N \times  d_{h}$, where the middle dimension of the high-rank feature satisfies $d_{h} \geq d$ (e.g., $d_{h} = 128$). Next, we use the down-projection layer with parameters $W^h_{down} \in \mathbb{R}^{d_h \times d}$ to restore the feature dimensionality from $B \times N \times d_h$ to $B \times N \times d$, where B is the batch size and N is the sequence length.
>
> For the relationship between ViDAs and the original network, with identical inputs and no non-linear operations, the three branches exhibit a linear relationship. For example, let Wv and Wo denote the parameter matrices for ViDA and the linear layer of the original model, respectively. Since they share the same input x, their outputs can be expressed as Y1 = Wv ⋅ x and Y2 = Wo ⋅ x, where ⋅ denotes matrix multiplication. Combining these outputs yields: Y = Y1 + Y2 = (Wv + Wo) ⋅ x. It is evident that this combination remains linear property, involving only matrix multiplication and addition operations without introducing non-linear transformations.

---

> ### Author Response · Authors · 2023-11-17
> **Rebuttal response to Reviewer Ut2z (Q2)**
>
> **Q2. The parameter 𝜭 in equation 5**
>
> 𝜭 is an empirical parameter employed to regulate the adjustment of $\lambda_{h}$ and $\lambda_{l}$ within the Homeostatic Knowledge Allotment (HKA) strategy. Utilizing statistical measures, we have computed an average uncertainty value of approximately 0.2. Consequently, we configure 𝜭 to be 0.2 to enhance the discrimination between samples exhibiting substantial distribution shifts and those with minor distribution shifts. We will add this detailed explanation in the revised version.

---

> ### Author Response · Authors · 2023-11-17
> **Rebuttal response to Reviewer Ut2z (Q3)**
>
> **Q3. incorporating the details from the appendix to the main paper.**
>
> Thank you for your valuable comments. We will move the ablation study on HKA from the appendix to the main paper. We have made modifications in the **updated PDF paper, specifically in Table 6 and Section 4.6, both marked in blue font**.

---

> ### Author Response · Authors · 2023-11-21
> **Further discussion to Reviewer Ut2z**
>
> Dear Reviewer Ut2z,
>
> As the discussion phase is quickly passing, we would like to know if you have any further questions or suggestions. We are more than happy to discuss. Thanks again for your valuable reviews!
>
> Best regards,
>
> All Anonymous Authors

---

> > ### Comment · Reviewer_Ut2z · 2023-11-22
> >
> > Thanks to the Authors for answering my questions.
> >
> > I confirm my initial positive evaluation of the paper.

---

> > > ### Author Response · Authors · 2023-11-23
> > >
> > > Dear Reviewer Ut2z,
> > >
> > > Thank you for endorsing our paper, and we appreciate your valuable time and constructive comments.
> > >
> > > Best regards,
> > >
> > > All Anonymous Authors

---

### Official Review · Reviewer_4MXU · 2023-11-01

**Soundness:** 3 good
**Presentation:** 3 good
**Contribution:** 3 good
**Rating:** 6
**Confidence:** 4

**Summary:**

The paper proposes to add two branches of low-rank and high-rank adapters to handle the problem of continuous test time adaption. The paper suggests that the low-rank adapter acquires domain-agnostic information, whereas the high-rank adapter acquires domain-specific knowledge. In addition, the paper proposes a Homeostatic Knowledge Allotment (HKA) technique for determining the contribution of the two branches. Experiments are carried out for the image classification task on the ImagenetC, CIFAR10C, and CIFAR100C benchmarks, as well as for semantic segmentation on the Cityscapes-to-ACDC benchmark. The experimental findings show that the proposed mechanism is successful, producing state-of-the-art outcomes for some pre-trained deep neural network architectures.

**Strengths:**

1. This paper employs two distinct adapters for acquiring domain-specific and domain-agnostic knowledge: low-rank and high-rank adapters.
2. The HKA technique for updating the weight contribution of these adapters depending on the prediction uncertainty value is a good idea.
3. The contribution of several components is demonstrated in the ablation analysis.
4. In the TTA scenario, experimental results for zero-shot generalization are novel.

**Weaknesses:**

1. There is no theoretical justification or an intuitive explanation for why the low-rank adapter acquires domain-agnostic information, whereas the high-rank adapter collects domain-specific knowledge other than experimental observation.
2. The backbone architecture in some experiments differs from prior developments, such as CoTTA.
3. What about the performance of the proposed approach CIFAR100-to-CIFAR100C method for ResNeXt-29 architecture?
4. Using the source data, retrain the "model added with low/high-rank adapters" for a few stages. Without access to the source domain data, it is impossible to employ off-the-shelf pre-trained models.

**Questions:**

1. There is no theoretical justification, nor even an intuitive explanation, for why the low-rank adapter acquires domain-agnostic information whereas the high-rank adapter collects domain-specific knowledge other than experimental observation.
2. The backbone architecture in some experiments differs from prior developments, such as CoTTA.
3. What about the performance of the proposed approach CIFAR100-to-CIFAR100C method for ResNeXt-29 architecture?
4. Using the source data, retrain the "model added with low/high-rank adapters" for a few steps. Without access to the source domain data, it is impossible to employ off-the-shelf pre-trained models.

---

> ### Author Response · Authors · 2023-11-17
> **Rebuttal response to Reviewer 4MXU (Q1)**
>
> **Q1. More intuitions of the low-rank adapter and high-rank adapter**
>
> Thank you for the constructive advice. To further validate the distinct domain representations of low-rank and high-rank ViDA , we conduct an additional experiment to invert the scale factors within the Homeostatic Knowledge Allotment (HKA) strategy. Specifically, for samples exhibiting high uncertainty, we reduce $\lambda_{h}$ while increasing $\lambda_{l}$. As shown in the Table below and Table 12 of the Appendix, building upon the ablation study of the submission, we integrate the **Inversed HKA** approach into Ex4, which already incorporates both low-rank and high-rank ViDAs. This adaptation yields an average error rate of 46.3 on ImageNet-C, marking a slight 0.7% and 2.9% error rate increase compared to Ex4 and Ex5, respectively. This experiment underscores how the proposed HKA strategy promotes the different domain representations between low-rank and high-rank VIDA models.
>
> |   | Contributions |   Average error rate |
> | --- | --- | --- |
> | Ex4 | ViDAh + ViDAl | 45.6 |
> | Ex5 | ViDAh + ViDAl + HKA | 43.4 |
> | **Inversed HKA** | ViDAh + ViDAl + Inversed HKA | **46.3** |
>
> To provide more straightforward verification, we extend our analysis by incorporating the qualitative analysis of Class Activation Mapping (CAM) in Figure 4 of the submission. Besides, we supplement and visualize the CAM of our proposed ViDAs and Inversed HKA in **Figure 4 of the updated PDF paper**. Meanwhile, we incorporate relevant explanations, marked in blue font. The low-rank ViDA is inclined to put more weight on the foreground sample while tending to disregard background noise shifts. This indicates that the low-rank ViDA attends to locations with more general and task-relevant information. The high-rank ViDA allocates more attention to locations characterized by substantial domain shift, encompassing the entirety of the input images. This behavior aligns with the high-rank branch's tendency to fit global information and predominantly extract domain-specific knowledge from the target domain data. Our proposed ViDAs exhibit higher response values on foreground samples, showcasing improved feature representation in target domains. Conversely, the Inversed HKA displays inferior feature attention, concentrating solely on specific foreground samples and failing to encompass complete semantic information. These visualizations, combined with previous results, jointly validate the intuition that the high-rank adapter focuses on domain-specific knowledge, while the low-rank adapter emphasizes domain-shared knowledge.

---

> ### Author Response · Authors · 2023-11-17
> **Rebuttal response to Reviewer 4MXU (Q2)**
>
> **Q2. CIFAR100-to-CIFAR100C experiment with ResNeXt-29 architecture**
>
> Thank you for your comprehensive feedback. The following table presents the results of our CIFAR100-to-CIFAR100C CTTA experiment with ResNeXt-29 as the backbone. Our approach substantially reduces the error rate by 15.7% compared to the source model and achieves competitive performance compared with other CTTA methods. We have supplemented the table and incorporated relevant explanations in **Appendix Section C.5, marked in blue font**.
> Combining Table 1 and Table 13 in the submission, we have validated the effectiveness of our method on three classification CTTA benchmarks using the same CNN backbones as CoTTA (Wang et al., 2022).
>
> | Method(ResNeXt-29) |   Source | BN adapt  | Tent  | CoTTA | Ours |
> | --- | --- | --- | --- | --- | --- |
> | Average error rate |   46.4 | 35.4 | 60.9 | 32.5 | 30.7(+15.7%) |

---

> ### Author Response · Authors · 2023-11-17
> **Rebuttal response to Reviewer 4MXU (Q3)**
>
> **Q3. Pre-train on source data**
>
> Pre-training the low-rank and high-rank ViDAs using source data is an unnecessary step and does not compromise the effectiveness of our approach. ViDAs can demonstrate comparable CTTA performance when they have a relatively stable initial parameter. As illustrated in the following table, we conducted an additional experiment on the Cityscape-to-ACDC scenario. ViDAs with random initial parameters and ViDAs with parameters pre-trained on ImageNet achieved 60.5 and 61.4 mIoU in target domains, respectively, exhibiting notable improvements compared to previous methods.
> To better showcase the practicality, we have included the aforementioned experiments with other parameter initialization techniques in **Appendix C.4, along with corresponding analyses highlighted in blue font**.
>
> |  | Adapter pretrain | Fog | Night | Rain | Snow | Mean (IoU) |
> | --- | --- | --- | --- | --- | --- | --- |
> | Source [58] | - | 69.1 | 40.3 | 59.7 | 57.8 | 56.7 |
> | CoTTA [57] | - | 70.9 | 41.2 | 62.4 | 59.7 | 58.6 |
> | Ours | Source | **71.6** | 43.2 | **66.0** | 63.4 | 61.1 |
> | Ours | Random initial | **71.6** | 43.6 | 64.9 | 61.9 | 60.5 |
> | Ours | ImageNet | **71.6** | **44.3** | **66.0** | **63.5** | **61.4** |

---

> ### Author Response · Authors · 2023-11-22
> **Further discussion to Reviewer 4MXU**
>
> Dear Reviewer 4MXU,
>
> Thanks again for your great efforts and constructive comments in reviewing this paper! As the discussion period comes to an end, we are eagerly anticipating your feedback and thoughts on our response. We sincerely hope you will consider our reply in your assessment. If there are any unclear explanations or remaining concerns, rest assured that we are committed to promptly resolving them and providing you with a comprehensive response.
>
> Best regards,
>
> All Anonymous Authors

---

> > ### Comment · Reviewer_4MXU · 2023-11-22
> > **Acknowledgement to Authors' Response and an additional query**
> >
> > Thanks for the additional experimental results and analysis. Most of my queries have been responded to, with experiments for some benchmarks.
> >
> > I have another question:
> > Do you have a comparison in terms of parameter efficiency due to the addition of adapters and freezing of the rest of the network parameters?

---

> > > ### Author Response · Authors · 2023-11-22
> > > **Further response to Reviewer 4MXU's comments**
> > >
> > > Thank you for your valuable feedback. As shown in the following table, the updated ViDA parameters can be reduced to 7.6% of the original model parameters, significantly lowering computational costs in the continual adaptation process. Meanwhile, since there is a linear relationship between the linear layer of the original model and our injected ViDAs, ViDAs can be projected into the original model through re-parameterization (Ding et al., 2021) during inference. This ensures that there is no additional increase in total model parameters, maintaining the plasticity of the original model.
> > >
> > > | Method  |   Our ViDAs | Original model  |
> > > | --- | --- | --- |
> > > | Parameter(MB) |   25.1 | 330.3  |

---

> ### Author Response · Authors · 2023-11-23
>
> Thanks again for your valuable time and feedback. I would like to inquire, can our response address your new query? If you have any additional questions, we would be happy to address them. If your concerns have been addressed, we would greatly appreciate it if you would consider raising your score.

---

### Official Review · Reviewer_RZSt · 2023-11-01

**Soundness:** 2 fair
**Presentation:** 3 good
**Contribution:** 2 fair
**Rating:** 6
**Confidence:** 4

**Summary:**

This paper studies the problem of continual test-time model adaptation, where training data is not accessible and only continually changing target domains are available. To address error accumulation and catastrophic forgetting problems, they propose a homeostatic Visual Domain Adapter (ViDA) which explicitly manages domain-specific and task-relevant knowledge. Moreover, a Homeostatic Knowledge Allotment (HKA) strategy is introduced to dynamically merge knowledge from low-rank and high-rank ViDAs. Extensive experiments on standard TTA benchmarks demonstrate the effectiveness of the proposed approach.

**Strengths:**

- The problem of tackling continuously shifting target domains represents a substantial challenge and is an area that has not been sufficiently explored in existing literature.

- The paper is articulately composed and comprehensible, effectively communicating the core ideas of the study to the reader. I am confident that I have acquired a solid understanding of the authors’ work through my examination of the manuscript.

- Experiments are comprehensive with a variety of benchmark tasks.

**Weaknesses:**

- My main concern is the novelty. Separating the extracted features into domain-invariant and domain-specific components has been extensively explored in previous domain adaptation and domain generalization methods. It lacks clear pieces of evidence to show why the proposed method is preferable in the context of test-time adaptation.

- The experimental results, as illustrated in Table 1, indicate that the improvements achieved by the proposed method in comparison to previous approaches, such as CoTTA, are marginal.

- The section of the paper discussing large models, as well as the associated experiments, appear incomplete and somewhat abrupt.

**Questions:**

Please refer to the weaknesses.

---

> ### Author Response · Authors · 2023-11-17
> **Rebuttal response to Reviewer RZSt (Q1)**
>
> **Q1. Clear pieces of evidence**
>
> Thank you for your patient response. First, we would like to emphasize the novelty and unique design of our method for the Continual Test-Time Adaptation (CTTA) task. The CTTA task differs from previous domain adaptation (DA) and domain generalization (DG) tasks in that it requires simultaneously addressing both error accumulation and catastrophic forgetting problems while striking a balance between them. Classical DA methods mainly focus on aligning features between the source domain and target domain to extract domain-invariant knowledge. Meanwhile, classical DG methods utilize meta-learning or feature alignment schemes to absorb domain-invariant information from source data (Zhou et al., 2021; Li et al., 2017). However, for CTTA tasks, practical constraints prohibit access to source domain data. This significantly increases the difficulty of extracting various domain knowledge. To this end, we propose the Homeostatic low-rank and high-rank adapters to efficiently extract and manage both domain-specific and domain-shared knowledge, addressing the specific error accumulation and catastrophic forgetting problems in CTTA. Meanwhile, the CTTA task places a strong emphasis on adaptation efficiency since each sample is only observed once. Therefore, our proposed method enables the efficient extraction of different domain knowledge using efficient adapter parameters, without compromising the model plasticity. This stands in contrast to absorbing knowledge through cumbersome model parameters. To further address the CTTA challenges and improve its efficiency, we introduce the Homeostatic Knowledge Allotment (HKA) strategy based on the distribution of each sample. This strategy adaptively combines diverse domain knowledge from each ViDA, making the most effective use of every opportunity to encounter each target sample.
>
> To comprehensively showcase the effectiveness of the proposed method in the context of CTTA, we systematically validate each contribution step by step. Initially, we conduct an ImageNet-to-ImageNet-C CTTA experiment utilizing a combination of two high-rank adapters or two low-rank adapters, as elaborated in Appendix Table 7. To ensure fairness, these experiments were conducted without implementing the HKA strategy and with an equal number of adapters as our proposed ViDAs. Remarkably, the low-rank adapters (Ex2) consistently demonstrated lower long-term error rates compared to both the source model and the two high-rank adapters. This can be attributed to the inclination of the low-rank ViDAs to learn general information and alleviate catastrophic forgetting in the CTTA problem. On the other hand, the performance of the high-rank adapters initially surpassed our approach (Ex4) in the early stages, covering the first few target domains. However, a noticeable degradation in performance became evident in later target domains. This observation highlights a crucial finding: while high-rank ViDAs enhance domain-specific knowledge acquisition and efficient cross-domain learning during the initial phases of CTTA, they simultaneously exacerbate catastrophic forgetting throughout the entire adaptation process.
>
> Then, we validate the effectiveness of the HKA strategy in the CTTA task. We conduct an additional experiment to invert the scale factors within the HKA strategy. Specifically, for samples exhibiting high uncertainty, we reduce $\lambda_{h}$ of the high-rank adapter while increasing $\lambda_{l}$ of the low-rank adapter, employing a strategy opposite to that in the submission. As shown in the following table, building upon the ablation study (Table 6) of the main paper, we integrate the Inversed HKA approach into Ex4, which already incorporates both low-rank and high-rank ViDAs. This adaptation yielded an average error rate of 46.3 on ImageNet-C, marking a 2.9% error rate increase compared to Ex5. This experiment highlights how the proposed HKA strategy promotes different domain representations between low-rank and high-rank VIDAs and can more efficiently address the CTTA task.
> |   | Contributions |   Average error rate |
> | --- | --- | --- |
> | Ex4 | ViDAh + ViDAl | 45.6 |
> | Ex5 | ViDAh + ViDAl + HKA | 43.4 |
> | **Inversed HKA** | ViDAh + ViDAl + Inversed HKA | **46.3** |

---

> ### Author Response · Authors · 2023-11-17
> **Rebuttal response to Reviewer RZSt (Q2)**
>
> **Q2. The experiment results of Table 1**
>
> First, as shown in Table 1 of the submission, our approach achieves an 11.4% improvement in classification accuracy compared to CoTTA when ViT-base is used as the backbone. These results indicate that our approach is more effective when ViT is employed as the backbone, leveraging its enhanced global context awareness. ViT, with its powerful self-attention mechanism, can adaptively comprehend the domain-specific and domain-shared knowledge extracted by our low-rank and high-rank ViDA.
>
> Second, utilizing ResNet50 as the backbone, our method outperforms the CoTTA approach by 1.5% in classification accuracy. Although the performance enhancement may not be as remarkable as observed with ViT as the backbone, other methods applied to ResNet50 also demonstrate marginal improvements. Meanwhile, our method is also a memory-efficient CTTA approach. In contrast to CoTTA, the updated ViDA parameters can be reduced to 7.6% of the updated CoTTA parameters, significantly reducing computational costs in the continual adaptation process.

---

> ### Author Response · Authors · 2023-11-17
> **Rebuttal response to Reviewer RZSt (Q3)**
>
> **Q3. Large model experiments**
>
> Thank you for your comprehensive comments. Examination of Tables 2 and 3 in the submission reveals a significant increase in the average error rate, from 28.2% to 39.3%, with the pre-trained encoder parameters of SAM. This is attributed to SAM, being a pixel-level foundational model, encountering limitations when transferred to image-level classification tasks.
> To support this observation, we conducted segmentation CTTA using SAM pre-trained parameters on the Cityscapes-to-ACDC scenario. It's worth noting that Segformer (Xie et al., 2021), utilized in the primary experiments (Table 4 of the submission), lacks positional encoding. Therefore, we adopt the SETR model (Zheng et al., 2021) as our new baseline, incorporating SAM's pre-trained parameters. As shown in the table below or Table 9 of the Appendix, our approach with SAM pre-trained parameters outperforms other methods in the ACDC target domains. This affirms our hypothesis: SAM, functioning as a pixel-level foundational model, excels in capturing fine-grained feature representations in dense CTTA tasks. We plan to further refine experiments with larger models in the revised version.
>
> |  | Pretrained | Fog | Night | Rain | Snow | Mean (IoU) |
> | --- | --- | --- | --- | --- | --- | --- |
> | Source | Source model | 	72.6|	43.1|	63.0|	64.3|	60.8|
> | Source | SAM | 74.8|	44.1|	66.7|	66.6|	63.0 |
> | Cotta | SAM | 75.4|	45.9|	67.3|	68.7|	64.3|
> | Ours | SAM | **76.5** | **47.2** | **68.1** | **70.7** | **65.6** |

---

> ### Author Response · Authors · 2023-11-23
> **Further discussion to Reviewer RZSt**
>
> Dear Reviewer RZSt,
>
> Thanks again for your great efforts and constructive comments in reviewing this paper! As the discussion phase will close in a few hours, we eagerly anticipate your feedback and thoughts on our response. We sincerely hope you will consider our reply in your assessment. If there are any unclear explanations or remaining concerns, rest assured that we are committed to promptly resolving them and providing you with a comprehensive response.
>
> Best regards,
>
> All Anonymous Authors

---

### Official Review · Reviewer_8EgD · 2023-11-03

**Soundness:** 3 good
**Presentation:** 3 good
**Contribution:** 3 good
**Rating:** 6
**Confidence:** 4

**Summary:**

The paper introduces a Continual Test-Time Adaptation (CTTA) method called Visual Domain Adapter (ViDA), which addresses the challenges of error accumulation and catastrophic forgetting in models operating in dynamic environments. ViDA differentiates itself by leveraging both high-rank and low-rank feature spaces to maintain domain-specific and shared knowledge. The HKA(Homeostatic Knowledge Allotment) is also proposed to balance the integration of these features dynamically. This proposed method is evaluated on multiple benchmarks, showing good performance in classification and segmentation tasks.

**Strengths:**

The paper presents an intriguing exploration of the differential capabilities of low-rank and high-rank features in domain adaptation, suggesting potential for a wide range of applications. The analysis substantiates the underlying mechanisms of the proposed method's operation. Also, the proposed method is validated through extensive experiments. The experiments include several benchmarks in classification and segmentation tasks.

**Weaknesses:**

- Observing the t-SNE plots in the low-rank branch (Figure 1), it appears that the nighttime representation in the ACDC dataset is not aligning as a domain-shared characteristic. I think nighttime data is not clustering in the low-rank space, which may indicate unique domain characteristics. Quantitative analysis with clustering metrics could clarify this.

- In Figure 3(a), the inter-domain divergence does not seem to exhibit significant differences between high-rank and low-rank features across conditions c1 to c9. Does this imply that the effectiveness of the rank-based approach is sensitive to the type of corruption present in the data? Understanding this could inform the robustness of the proposed method across a wider array of scenarios.

**Questions:**

- Regarding Figure 1, it would be insightful to include the t-SNE results for each transformer block. Such detailed visualizations would help to understand feature spaces across the different layers of the network.

- Since t-SNE visualizations represent relative rather than absolute distances, I am curious about the quantitative differences in domain distribution distances within the ACDC dataset when comparing low-rank to high-rank representations. Quantifying this difference could bolster the demonstration of your method's efficacy in managing domain shifts.

---

> ### Author Response · Authors · 2023-11-17
> **Rebuttal response to Reviewer 8EgD (Q1)**
>
> **Q1. Quantitative analysis of domain distribution distances within the ACDC dataset**
>
> Thank you for the constructive advice. We conduct a quantitative analysis of the domain distribution distances within the ACDC dataset when comparing low-rank and high-rank representations. Specifically, following the k-means clustering metrics (MacQueen, 1967), we first calculate the Euclidean distance clustering center in every target domain by considering all samples within each domain.Therefore, in the ACDC dataset, we obtain four domain clustering centers and use these centers to represent each target domain distribution. Using the same calculation process, we compute the distances for the source model, low-rank branch, and high-rank branch, utilizing features from the third transformer block.The quantitatively normalized results are presented in the table below, showing in the order of continual adaptation (Fog → Night → Rain → Snow).
>
> | **Distance** | **Source model** | **Low-rank branch** | **High-rank branch** |
> | --- | --- | --- | --- |
> | **Fog-Night** | **0.73** | **0.29** | **0.98** |
> | **Night-Rain** | **0.62** | **0.23** | **0.65** |
> | **Rain-Snow** | **0.51** | **0.10** | **0.80** |
> | **Snow-Fog** | **0.38** | **0.13** | **0.47** |
> | **Mean** | **0.56** | **0.19** | **0.73** |
>
> Based on the experimental results, we observe that the Low-rank adapter significantly reduces the domain distribution distance in the ACDC dataset. Especially in the Fog-Night and Night-Rain scenarios, the distance decreased from 0.73 to 0.29 and from 0.62 to 0.23, respectively. Compared to the Source model, the Low-rank adapter noticeably aligns the feature distribution between different domains, showing a relatively domain-shared representation. In the case of the high-rank adapter, there is a larger distance between domains. In conjunction with the lower intra-class divergence presented in Figure 3(b) of the submission, it is evident that the high-rank adapter effectively facilitates domain-specific knowledge extraction in each target domain.

---

> ### Author Response · Authors · 2023-11-17
> **Rebuttal response to Reviewer 8EgD (Q2)**
>
> **Q2. The factors influencing the effectiveness of the rank-based approach.**
>
> In Continual Test-Time Adaptation (CTTA), two main factors influence the effectiveness of low-rank and high-rank: **1)** different target domain distributions (types of corruption), and **2)** the length of continual adaptation. To explore the impact of corruption types, as demonstrated in Appendix Table 7, we conduct an ImageNet-to-ImageNet-C CTTA experiment using either high-rank adapters or low-rank adapters. To ensure fairness, these experiments were carried out without implementing the Homeostatic Knowledge Allotment (HKA) strategy and employing an equal number of adapters as our proposed ViDAs. Across all corruption types, both the low-rank adapters (Ex2), high-rank adapter (Ex3), and our ViDAs (Ex4) outperformed the baseline source model's classification accuracy (Ex1), confirming the robustness and effectiveness of our contributions. However, for various target domains, the classification capability and feature representation of low-rank and high-rank adapters indeed differ. For instance, there is a significant difference among them regarding major target corruptions such as brightness, snow, frost, fog, and contrast, whereas the distinction is less pronounced for a few target corruptions like pixelate and Jpeg.
>
> On the other hand, the impact on the effectiveness of the rank-based approach also arises from the length of continual adaptation. As illustrated in Appendix Figure 5 (b), we conducted a 10-round CTTA experiment on ImageNet-to-ImageNet-C, simulating a long-term adaptation scenario by repeating 10 rounds of 15 corruption sequences in ImageNet-C. Remarkably, the high-rank ViDA achieves competitive results compared to other methods during the initial 1 to 3 rounds, demonstrating the capacity of high-rank features to efficiently learn target domain-specific knowledge. However, an increase in error rates becomes evident during the later rounds (rounds 5 to 10). These results validate the potential for encountering catastrophic forgetting when exclusively focusing on domain-specific knowledge. In contrast, the performance of the low-rank ViDA remains consistently robust throughout the continual adaptation process, verifying it concentrates more on extracting domain-shared knowledge and effectively preventing the catastrophic forgetting problem. This highlights the distinct effectiveness of the two types of adapters at different continual adaptation stages.
>
> Finally, as shown in the table below, we extend the inter-domain divergence in Figure 3 of the submission. In the first round, the average inter-domain divergence for high-rank adapters is 0.34, whereas for low-rank adapters, it is 0.30. Moving to the second round, the high-rank adapter and low-rank adapter achieve average inter-domain divergences of 0.37 and 0.29, respectively. In summary, with the increase in the length of continual adaptation, the gap in inter-domain divergence between high-rank and low-rank features also widens.
>
> | **Inter-Domain Div (mean)** | **Low-rank adapter** | **High-rank adapter** |
> | --- | --- | --- |
> | **1 Round** | **0.30** | **0.34** |
> | **2 Round** | **0.29** | **0.37** |
> | **3 Round** | **0.30** | **0.42** |
>
> In conclusion, the absence of notable inter-domain divergence among conditions c1 to c9 can be ascribed to the combined influence of the corruption types and their early stage in continual adaptation.

---

> ### Author Response · Authors · 2023-11-17
> **Rebuttal response to Reviewer 8EgD (Q3)**
>
> **Q3. The t-SNE results for each transformer block**
>
> We supplement and visualize the t-SNE results for each transformer block in **Appendix Figure 5 (a) of the updated PDF paper.** Meanwhile, we incorporate corresponding explanations, marked in blue font. The visualization revealed that the qualitative results obtained from different layers (i.e., transformer blocks 1, 2, and 4) of the Segformer-B5 model exhibit similar distribution representations. In detail, the low-rank ViDA effectively reduces the distribution distance across different target domains, indicating its focus on extracting task-relevant knowledge. On the other hand, the high-rank ViDA exhibits notable distribution discrepancies among the various target domains, indicating its focus on extracting domain-specific knowledge.

---

> ### Author Response · Authors · 2023-11-23
> **Further discussion to Reviewer 8EgD**
>
> Dear Reviewer 8EgD,
>
> Thank you once again for your great efforts and constructive comments in reviewing this paper! As the discussion phase will close in a few hours, we eagerly await your feedback and thoughts on our response. We sincerely hope you will consider our reply in your assessment. If there are any unclear explanations or remaining concerns, rest assured that we are committed to promptly resolving them and providing you with a comprehensive response.
>
> Best regards,
>
> All Anonymous Authors

---

### Meta-Review · Area_Chair_sidW · 2023-12-05

**Metareview:**

Dear authors,

All the reviewers have ranked the draft positively, and at the same time, improvements have been suggested. Please include those in your final draft.

One of the concerns is there is no theoretical proof of the claims that " low-rank adapter’s ability to learn long-term domain-shared knowledge".  Readability needs to be improved, for example, inverse HKA is not properly defined (slightly defined above conclusion). A minor point from the Meta-reviewer is the use of "tactfully design", it does not tell much about the algorithm or method being proposed. Please find more appropriate wording.

The authors appear to have used extensive line-space management, leading to a hard-to-read draft. Fig. 4 is overlapping text above it.

Authors are encouraged to update the draft by incorporating suggestions by the reviewers and also improve readability.
The final camera-ready version should not incorporate space management outside standard recommendations.



regards
Meta reviewer

**Justification For Why Not Higher Score:**

Writing could be improved. Claims do not have theoretical backing.

**Justification For Why Not Lower Score:**

Results are better and idea is interesting.

---

### Decision · Program_Chairs · 2024-01-16

Accept (poster)